# The long non-coding RNA HOTAIR contributes to joint-specific gene expression in rheumatoid arthritis

Muriel Elhai [1], Raphael Micheroli [1], Miranda Houtman[1], Masoumeh Mirrahimi[1], Larissa Moser[1], Chantal Pauli[2], Kristina Bürki[1], Andrea Laimbacher[1], Gabriela Kania [1], Kerstin Klein [1,3,4], Philipp Schätzle[5], Mojca Frank Bertoncelj[1], Sam G. Edalat[1], Leandra Keusch[1], Alexandra Khmelevskaya[1], Melpomeni Toitou [1], Celina Geiss [1], Thomas Rauer [6], Maria Sakkou[7,8], George Kollias [7,8], Marietta Armaka [9], Oliver Distler [1] & Caroline Ospelt [1] ✉

Although patients with rheumatoid arthritis (RA) typically exhibit symmetrical joint involvement, some patients develop alternative disease patterns in response to treatment, suggesting that different molecular mechanism may underlie disease progression depending on joint location. Here, we identify joint-specific changes in RA synovium and synovial fibroblasts (SF) between knee and hand joints. We show that the long non-coding RNA HOTAIR, which is only expressed in knee SF, regulates more than 50% of this site-specific gene expression in SF. HOTAIR is downregulated after stimulation with pro-inflammatory cytokines and is expressed at lower levels in knee samples from patients with RA, compared with osteoarthritis. Knockdown of HOTAIR in knee SF increases PI-Akt signalling and IL-6 production, but reduces Wnt signalling. Silencing HOTAIR inhibits the migratory function of SF, decreases SF-mediated osteoclastogenesis, and increases the recruitment of B cells by SF. We propose that HOTAIR is an important epigenetic factor in joint-specific gene expression in RA.

Chronic arthritis is a major public health problem, which has a substantial influence on health and quality of life[1]. Most forms of arthritis, including rheumatoid arthritis (RA), osteoarthritis (OA) and spondyloarthritis, have a distinctive topographical pattern of joint involvement[2]. Among them, RA is the most frequent autoimmune arthritis, affecting 1% of the population[3]. Despite the advances made in the management of RA in the last decades, 6–17% of the patients remain refractory to immunosuppressive treatment[4].

Although patients with untreated RA typically exhibit a symmetrical polyarthritis, individuals with refractory disease might develop a less extensive pattern of polyarthritis, an oligoarticular or even a monoarticular disease, suggesting that immunosuppressive therapy might be effective in some joints and not in others[5]. Thus, beyond differences between diseases, there are also differences in phenotype and response to treatment depending on the joints within the same type of arthritis,

[1]Center of Experimental Rheumatology, Department of Rheumatology, University Hospital of Zurich, University of Zurich, Zurich, Switzerland. [2]Institute for Pathology and Molecular Pathology, University Hospital Zurich, Zurich, Switzerland. [3]Department of BioMedical Research, University of Bern, Bern, Switzerland. [4]Department of Rheumatology and Immunology, Inselspital, Bern University Hospital, University of Bern, Bern, Switzerland. [5]Cytometry Facility, University of Zurich, Zurich, Switzerland. [6]Department of Trauma Surgery, University Hospital Zurich, Zurich, Switzerland. [7]Institute for Bioinnovation, Biomedical Sciences Research Center (BSRC) 'Alexander Fleming', Vari, Greece. [8]Department of Physiology, Medical School, National and Kapodistrian University of Athens, Athens, Greece. [9]Institute for Fundamental Biomedical Research, Biomedical Sciences Research Center "Alexander Fleming", Vari, Greece. ✉e-mail: caroline.ospelt@usz.ch

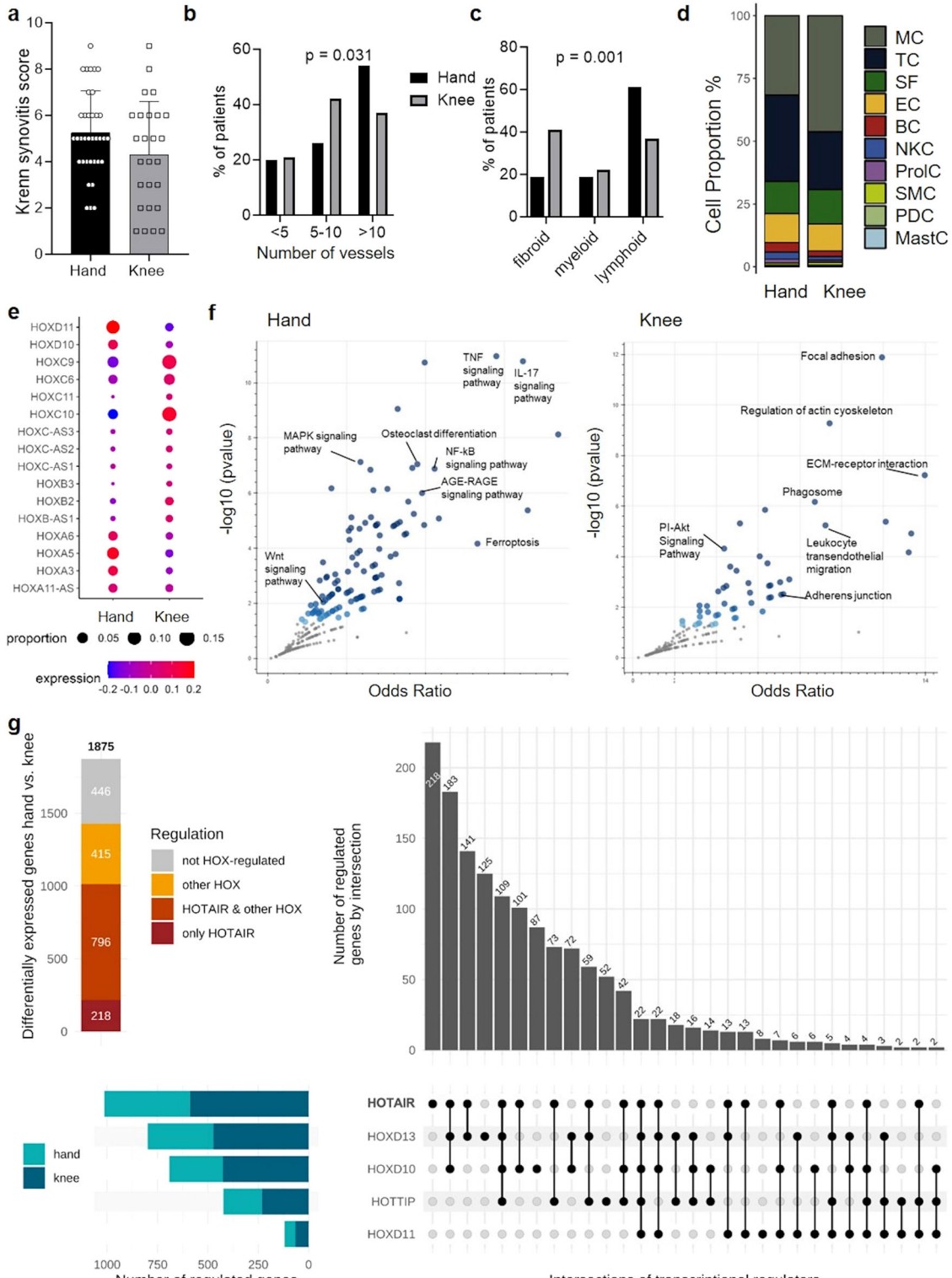

**Fig. 1 | *HOTAIR* changes site-specific gene expression in synovial fibroblasts.**
**a** Krenn synovitis score in hand (n = 36) and knee (n = 27) synovium from RA
patients. Mean +/− standard deviation is shown. **b** Vascularization as assessed by
CD31staining in synovium from RA patients (35 hands and 24 knees). Vessels were
counted in 5 fields (20x magnification). χ² test. **c** Synovial histological pattern in
hand (n = 36) and knee (n = 26) RA synovium, χ2 test. **d** Cell proportions between
hand (n = 8) and knee (n = 6) synovium using single cell RNA sequencing. SF
synovial fibroblasts, SMC smooth muscle cells, EC endothelial cells, MastC mast
cells, PC plasma cells, BC B cells, TC T cells, NKC NK cells, nGC neutrophilic gran-
ulocytes, MC myeloid cells, ProIC proliferating cells. **e** HOX gene expression in hand

(n = 8) and knee (n = 6) synovial fibroblasts from RA patients in single cell RNA
sequencing analysis. **f** Scatter dot plot of pathway enrichment analysis of genes
significantly enriched in hand SF (n = 8) and in knee SF (n = 4) (FDR < 0.05; Log fold
change +/−1). Blue dots: significantly enriched pathways, darker color corresponds
to lower *p*-values. Grey dots represent pathways with *p* > 0.05. Fisher exact test.
**g** Overlap between genes changed by *HOX* genes (*HOXD10*, *HOXD11*, *HOXD13*,
*HOTTIP*, *HOTAIR*) and genes differentially expressed between hand (n = 8) and knee
(n = 6) SF in single cell RNA sequencing. SF synovial fibroblasts. Not reported *p*-
values on the figure were not significant.

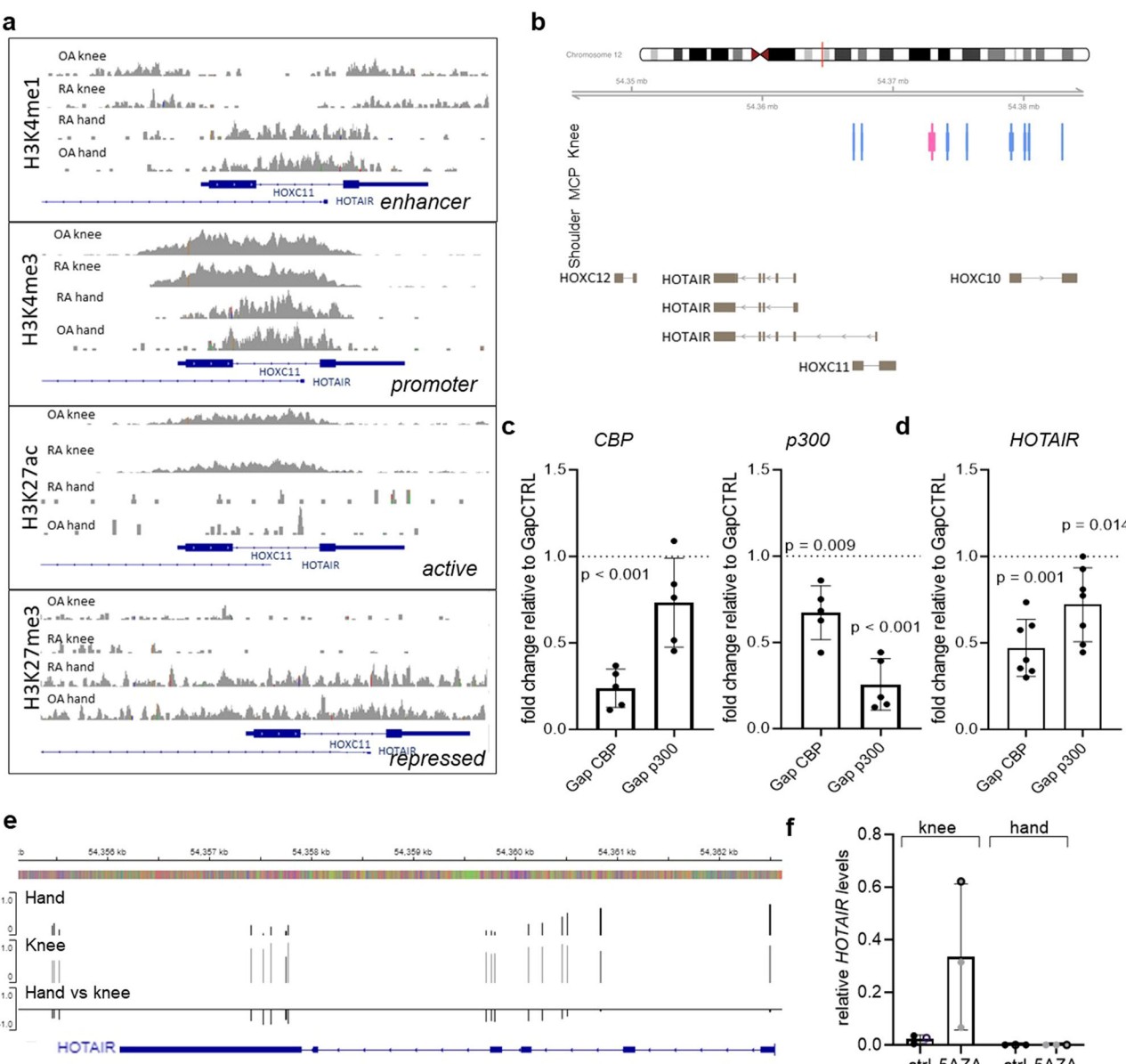

**Fig. 2 | Joint-specific *HOTAIR* expression is epigenetically imprinted.**
**a** Assessment of H3K4me1, H3K4me3, H3K27ac and H3K27me3 marks by ChIP seq over the promoter/transcription start site region of *HOTAIR*, which lies within the intron of HOXC11. Data was extracted from GSE163548. **b** CAGE (cap analysis of gene expression) of enhancers/promoters at the *HOTAIR* locus. Active promoters (light blue bars) and an enhancers (pink bars) are present in SF from knees ($n = 2$), but not in metacarpophalangeal (MCP) joints ($n = 3$) or shoulders ($n = 2$). Data were extracted from GSE163548. **c** Expression of *CBP* (left panel) and *p300* (right panel) was measured by qPCR in SF transfected with GapmeRs targeting *CBP* ($n = 5$) and *p300* ($n = 5$), respectively. Mean +/− standard deviation is shown. One sample *t* test.

**d** Expression of *HOTAIR* following silencing of SF for p300 and CBP, respectively ($n = 6$). Control transfected cells were set to 1. Mean +/− standard deviation is shown. One sample *t* test. **e** Average DNA methylation in hand ($n = 1$ OA/4 RA) and knee SF ($n = 1$ OA/4 RA) across the *HOTAIR* gene. Delta beta values (third row) show differentially methylated CpG sites between hand and knee samples. Human-Methylation 450 BeadChip data was taken from ref. 17. **f** Expression of *HOTAIR* following treatment with 1 μM 5-azacytidine (5AZA) for 5 days in knee and hand SF. Open circle RA SFs, closed circle OA SF ($n = 3$). Mean +/− standard deviation is shown. Not reported *p*-values on the figure were not significant.

suggesting that molecular mechanisms may differ according to joint location.

Deciphering the heterogeneity of synovium at both the cellular and molecular levels has revolutionized the understanding of the pathogenesis of RA[6–9]. In particular, refractory RA has been associated with a pauci-immune, fibroid histological pattern of the synovium and a molecular signature suggestive of activated fibroblasts[10]. Activation of synovial fibroblasts (SF) has long been known to play a critical role in joint inflammation and destruction[11,12], but has attracted considerable attention more recently due to the discovery of pathogenic

subpopulations of SF in the synovium through single-cell analysis[10,13–16]. Changes in the epigenetic landscape have been shown to be central to the permanent activation and aggressiveness of SF in RA[12].

We previously demonstrated the existence of transcriptomic, epigenetic and functional changes of SF depending on their joint location[17]. This specific stromal signature in particular concerned genes involved in the embryonic development of the respective joint regions (i.e. *HOX* genes). Joint specific patterns of DNA methylation in *HOX* genes were found in SF[18] as well as in cartilage[19] which suggested an embryonically imprinted joint specific stromal signature. In

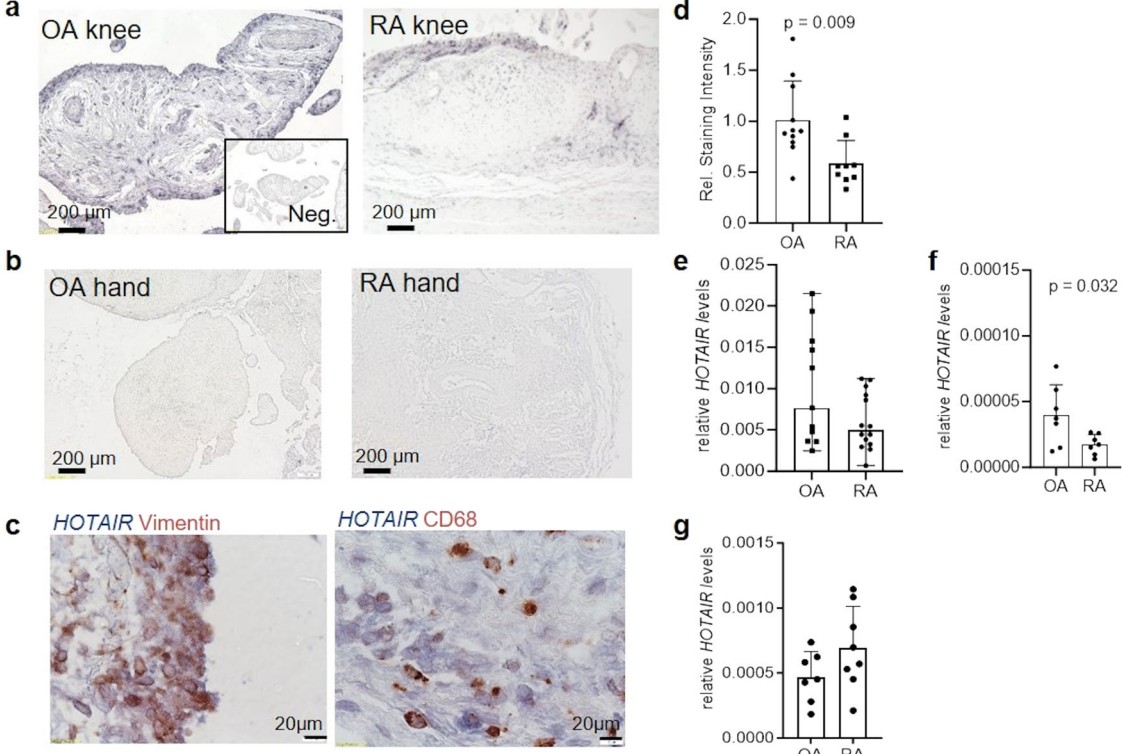

**Fig. 3 | *HOTAIR* is expressed in synovial tissues of lower extremity joints with higher expression in OA than in RA. a** Representative pictures of synovial tissues from knee joints of OA (left, *n* = 11) and RA (right, *n* = 9) patients stained for *HOTAIR* by in-situ hybridization (ISH). Magnification 100×. Inset shows staining with the anti-sense probe (negative control). **b** Representative picture of *HOTAIR* in-situ hybridization in synovial tissues of OA and RA hand joints (*n* = 3). Magnification 100×. **c** Double staining of *HOTAIR* (in blue) and vimentin (in red, left panel) or CD68 (in red, right panel) to assess *HOTAIR* expression in synovial fibroblasts and macrophages, respectively. Magnification 400×. **d** Relative quantification of *HOTAIR* ISH in OA (*n* = 11) and RA (*n* = 9) synovial tissue using ImageJ. Mean +/− standard deviation is shown. Unpaired *t* test. **e** Expression of *HOTAIR* measured by quantitative PCR in OA (*n* = 11) and RA (*n* = 14) synovial tissues normalized to *GAPDH* as housekeeping gene. Mean +/− standard deviation is shown. **f** Expression of *HOTAIR* measured by quantitative PCR in OA (*n* = 7) and RA (*n* = 7) synovial tissues normalized to *COL3A1*. Mean +/− standard deviation is shown. Unpaired *t* test. **g** Expression of *HOTAIR* measured by quantitative PCR in cultured OA (*n* = 7) and RA (*n* = 8) SF. Mean +/− standard deviation is shown. Unpaired *t* test. OA osteoarthritis, RA rheumatoid arthritis.

particular, the long non-coding RNA (lncRNA) HOX transcript anti-sense intergenic RNA (*HOTAIR*), which is an important regulator of the epigenetic landscape[20], was exclusively expressed in joints of the lower extremity in humans and in mice[17].

*HOX* genes encode a family of transcriptional regulators, which are involved in distinct developmental programmes along the head-tail axis of vertebrates[21,22]. During embryogenesis, the expression of specific *HOX* genes delineates distinct positional identities that lead to specific cell differentiation and tissue morphogenesis[20]. Interestingly, it has been shown that adult skin fibroblasts maintained key features of *HOX* gene expression patterns established during embryogenesis[23] and also in skin fibroblasts site-specific *HOX* expression is epigenetically maintained[20,24]. In addition to the skin, a site-specific expression of HOX genes is observed in several other differentiated tissues, e.g. in cartilage, the vasculature and gastrointestinal tract[25–29]. Nevertheless, it remains to be determined whether differences between the phenotype of arthritis at the tissue and molecular levels depend on its location, and if so whether the site-specific expression of *HOX* genes is involved in these changes.

Here, we identify joint-specific molecular and tissue changes in RA and show that *HOTAIR* (HOX transcript antisense RNA) modulates gene expression in SF in a joint-specific manner. Down-regulation of *HOTAIR* in an inflammatory environment led to activation of specific arthritis-relevant pathways and changes in SF function, which might modulate the arthritis phenotype in lower extremity joints.

## Results
### Joint-specific histological and molecular differences in RA
We first analysed joint-specific differences in the synovium of RA patients using a multi-level approach including histological and molecular analysis. Comparison of the histological grade of synovitis[30] between hand and knee RA showed a trend towards a higher synovitis score in RA hands (mean: 5.25 ± 1.78) compared to RA knees (mean: 4.31 ± 1.78; difference −0.94 ± 0.52, *p* = 0.076), as previously seen[17] (Fig. 1a). There was no association of the Krenn synovitis score with clinical characteristics of the disease in individual patients such as CRP or DAS28ESR (*p* = 0.71 and *p* = 0.48, respectively) (Supplementary Fig. S1a, b). Vascular density of the synovium was higher in RA hand synovium, with a larger percentage of patients having highly vascularised synovium in the hands than in the knees (54% vs. 37% *p* = 0.031) (Fig. 1b). Assessment of synovial histological patterns as defined by Humby et al.[9] in hand and knee RA showed a predominance of the lymphoid histological pattern in hand RA, whereas the histological patterns were balanced in the knees (Fig. 1c). The lymphoid histological pattern presents with strong infiltration of T and B cells in the synovium, the diffuse-myeloid histological pattern shows predominant influx of myeloid cells and the pauci-immune/fibroid histological pattern is characterised by scanty immune cells and prevalent stromal cells[9]. Consequently, analysis of synovial cell proportions by single cell RNA sequencing (scRNAseq) showed expansion of T- cell compartments in wrist compared to knee RA synovium and higher proportions of myeloid cells in knee versus wrist synovium (Fig. 1d,

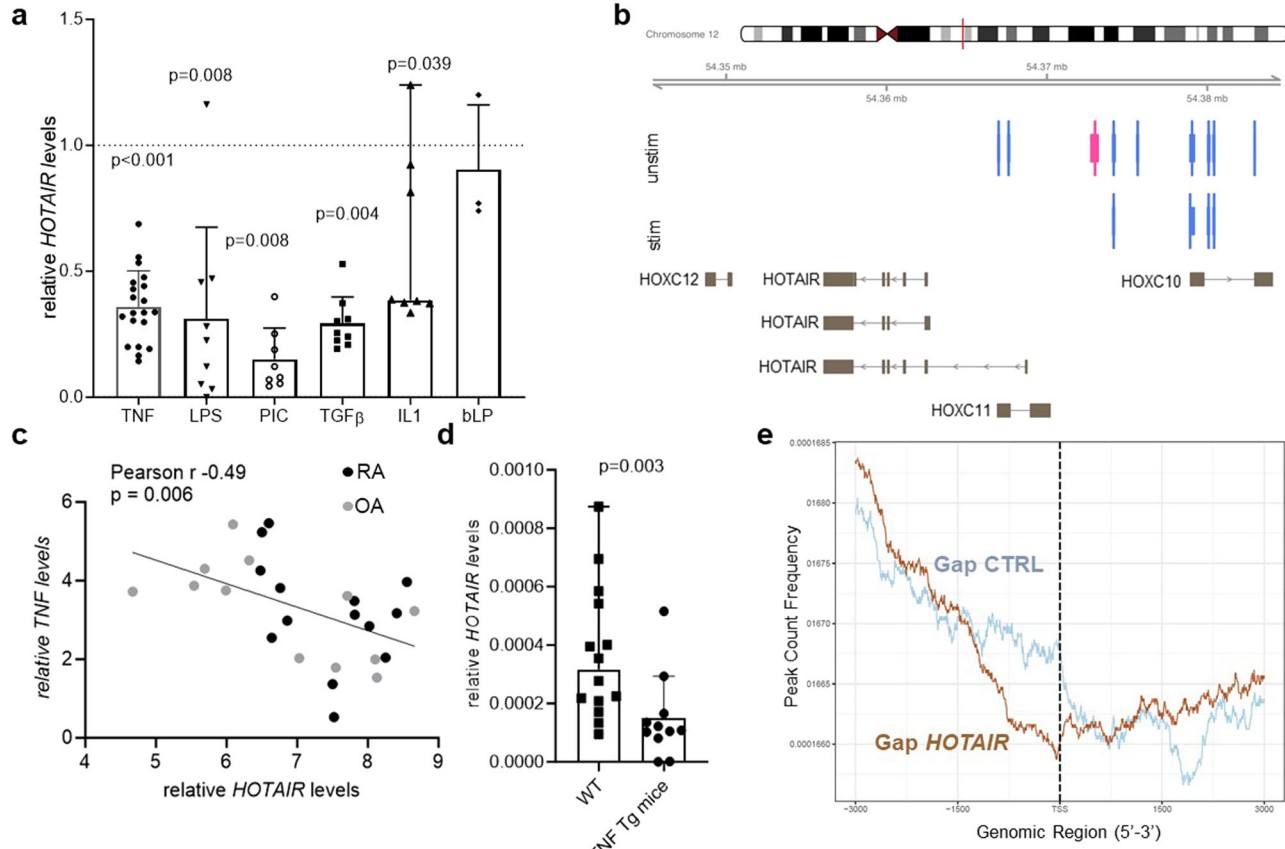

**Fig. 4 | *HOTAIR* is downregulated by inflammatory cytokines and shapes the epigenetic landscape of synovial fibroblasts. a** SF were left untreated or stimulated with TNF (*n* = 21), LPS (*n* = 9), poly I:C (PIC) (*n* = 9), TGFβ (*n* = 9), IL1β (*n* = 9), or bacterial lipoprotein (bLP) (*n* = 3) for 24 h. Expression of *HOTAIR* was measured by qPCR. Unstimulated samples were set to 1. Mean +/− standard deviation is shown, one sample *t* test (all but IL1β). IL1β: median and range, one sample Wilcoxon. **b** CAGE analysis of enhancers/promoters at the *HOTAIR* locus in SF from knees (*n* = 2) in basal conditions and after stimulation with TNF. After TNF stimulation several active promoters (light blue bars) and the enhancer (pink bar) disappear in knee SF. **c** Correlation between *TNF* and *HOTAIR* expression measured by qPCR in RA synovial tissues. Pearson correlation, one-sided. **d** Expression of *Hotair* was measured by qPCR in knee and ankle SF isolated from wildtype (WT) (*n* = 14) and TNF transgenic (Tg197) (*n* = 13) C57BL/6 mice. Median and range is shown. Mann–Whitney test. **e** ChIP sequencing of H3K27me3 marks in SF silenced for *HOTAIR* (*n* = 3) and control SF (*n* = 3) was performed 48 h after transfection. TSS Transcription start site. SF synovial fibroblasts, RA rheumatoid arthritis, OA osteoarthritis. Not reported *p*-values on the figure were not significant.

Supplementary Fig. S1c and Supplementary Table S1). In summary, all these data pointed towards higher inflammatory activity in hand versus knee RA synovium.

We then used the scRNAseq data to assess whether these site-specific tissue changes were associated with molecular changes in SF. In total, 1875 genes were differentially expressed in RA hand and knee SF, 834 genes were overexpressed in RA hand SF and 1041 in RA knee SF. We confirmed the joint-specific expression of HOX genes in SF in this dataset (Fig. 1e). In addition, various pathways that we had previously found to be differentially enriched in cultured hand and knee SF in vitro[17], were also joint-specifically activated in vivo, such as cell adhesion, extracellular matrix (ECM) interaction and bone remodeling/osteoclast differentiation pathways (Supplementary Data 1, 2, and Fig. 1f). Several enriched pathways were previously implicated to be relevant in RA such as MAPK, and Wnt and PI-Akt signaling[31]. Thus, these results confirmed joint specific gene expression in SF in terms of developmental as well as inflammatory pathways.

We then sought to understand in how far the joint specific expression of HOX transcripts was involved in these site-specific gene expression changes. To this end, we silenced SF for *HOXD10*, *HOXD11*, *HOXD13*, and the long non-coding RNAs *HOTAIR* and *HOTTIP*, respectively. These HOX transcripts were the most discriminating transcripts in cultured SF and synovial tissues between the hand and knee in our previous in vitro analysis[17]. Due to the lower sequence depths in

scRNAseq, the less expressed transcripts *HOXD13*, *HOTTIP* and *HOTAIR* were not detectable in the scRNAseq dataset (Fig. 1e). The differential gene expression by *HOX* gene silencing in vitro corresponded to 76.2% of the differential gene expression between the hand and knee in vivo using scRNAseq (Fig. 1g). Among the different *HOX* genes, *HOTAIR* alone or in combination with other *HOX* genes changed 54% of this joint-dependent gene expression. This suggested that joint-specific expressed HOX transcription factors and non-coding RNAs drive most of the joint-specific transcriptome in RA SF.

## Joint and disease specific expression of *HOTAIR*

We previously showed that *HOTAIR* is exclusively expressed in lower limb SF and joints in both mice and human[17]. We now further assessed the epigenetic mechanisms responsible for differential expression of *HOTAIR* between hand and knee SF. Analysis of the chromatin landscape of cultured RA and OA SF showed that hand SF carried repressive H3K27me3 marks broadly across the promoter and transcription start site of *HOTAIR* (Fig. 2a) and did not show any active promoter or enhancers as assessed by Cap Analysis of Gene Expression (CAGE-seq) (Fig. 2b). Knee SF used a different enhancer for *HOTAIR* expression (Fig. 2a), which was also shown to be active by CAGE-seq (Fig. 2b). Furthermore, knee SF had increased levels of activating H3K27ac marks in particular over the promoter region compared to hand SF (Fig. 2a). Accordingly, *HOTAIR* expression in cultured knee SF was

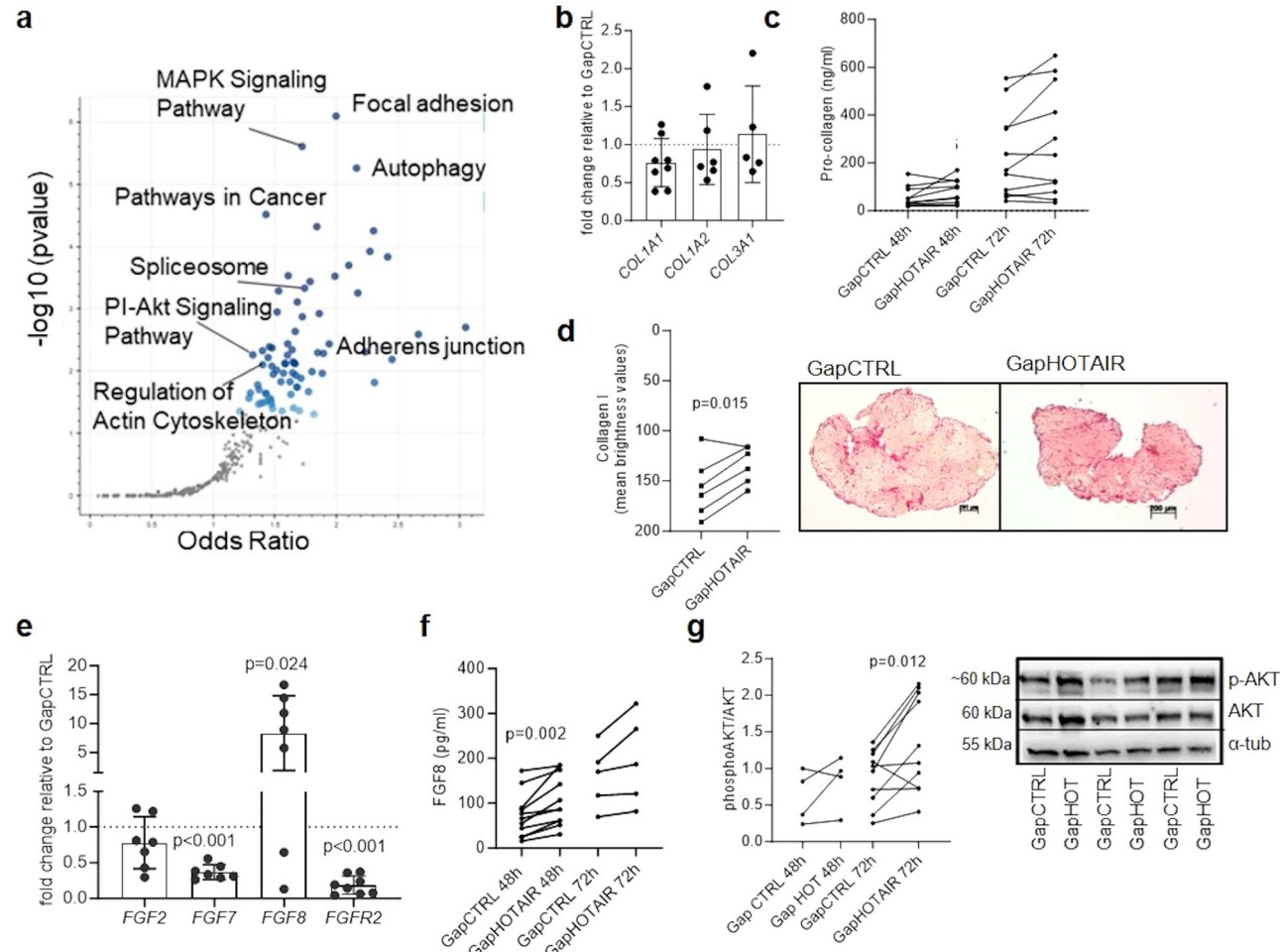

**Fig. 5 | *HOTAIR* modulates arthritis relevant pathways. a** Scatter dot plot of pathway enrichment analysis of genes significantly changed after *HOTAIR* silencing (FDR < 0.05; Log fold change +/− 1). Blue dots: significantly enriched pathways, darker color corresponds to lower *p*-values. Grey dots represent pathways with *p* > 0.05. Fisher exact test. **b** Expression of *COL1A1* (*n* = 8), *COL1A2* (*n* = 6) and *COL1A3* (*n* = 5) measured by quantitative PCR between SF transfected with control (GapCTRL) or *HOTAIR* targeting GapmeR (GapHOTAIR) after 48 h. Control transfected cells were set to 1. Mean +/− standard deviation is shown. **c** Pro-Collagen was measured in supernatants of SF transfected with control or *HOTAIR* targeting GapmeR after 48 h (*n* = 10) and 72 h (*n* = 11) by ELISA. **d** Collagen I was stained by immunohistochemistry in 3D micromasses formed with control or *HOTAIR* silenced

SF (*n* = 6). Right panel: representative pictures, 25× magnification, scale bare = 200 μm; left panel: quantification with ImageJ and analysis with paired *t* test. **e** Expression of *FGF2* (*n* = 7), *FGF7* (*n* = 8), *FGF8* (*n* = 7) and *FGFR2* (*n* = 8) measured by quantitative PCR between SF transfected with control or *HOTAIR* targeting GapmeR after 48 h. Control transfected cells were set to 1. Mean +/− standard deviation is shown. One sample *t* test. **f** FGF8 secretion in cell culture supernatants 48 h (*n* = 11) and 72 h (*n* = 5) after transfection measured by ELISA. Paired *t* test. **g** Expression of AKT and phosphorylated AKT in SF transfected with control or *HOTAIR* targeting GapmeR after 48 h (*n* = 4) and 72 h (*n* = 10). Right panel: representative examples; left panel: densitometric analysis. Paired *t* test.

decreased after silencing of the histone acetyltransferases p300 and CBP (Fig. 2c, d). In addition, we observed higher DNA methylation in the *HOTAIR* gene body in cultured SF from knees as compared to hands (Fig. 2e). DNA de-methylation by 5-azacytidine further induced *HOTAIR* expression in knee SF without changes in hand SF (Fig. 2f), suggesting that DNA methylation is an additional level of regulating *HOTAIR* gene expression in knee SF. Together these data confirm the existence of a strong, mitotically stable epigenetic imprinting of joint-specific *HOTAIR* expression in SF and suggest different levels of epigenetic control with histone modifications regulating joint-specific expression of *HOTAIR* and DNA methylation regulating *HOTAIR* expression in knee SF.

By in situ hybridisation (ISH), we confirmed that *HOTAIR* is expressed in synovial tissues from knees (Fig. 3a), but not from hands (Fig. 3b). In the synovium, *HOTAIR* was expressed mainly in SF, in both the lining and the sublining synovium (Fig. 3c). Analysis of *HOTAIR* expression in knee joints of OA and RA patients showed that *HOTAIR* was more abundant in OA than in RA knees (Fig. 3d, e). To ensure that

the difference observed was indeed linked to a difference in *HOTAIR* expression according to diseases and not to a difference in the number of fibroblasts in the tissue, we normalized the expression of *HOTAIR* in the tissue to a fibroblast specific gene, namely *COL3A1*. This approach did not change the result (Fig. 3f), confirming the difference in *HOTAIR* expression between OA and RA in vivo. However, the difference in *HOTAIR* expression between OA and RA was lost in cultured knee SF (*p* = 0.133) (Fig. 3g), suggesting that the lower expression of *HOTAIR* in RA joints was modulated by local factors in vivo.

To determine which local factors could influence the expression of *HOTAIR* in arthritis, we assessed *HOTAIR* expression in SF upon stimulation by various cytokines and Toll-like receptor (TLR) ligands. Stimulation with most of the inflammatory cytokines decreased *HOTAIR* expression in SF (Fig. 4a). Furthermore, active promoter and enhancer sites at the *HOTAIR* locus were closed after TNF stimulation (Fig. 4b). Consistently, *HOTAIR* expression inversely correlated with *TNF* expression in arthritic synovium (Fig. 4c). SF isolated from arthritic, TNF transgenic mice (TG197) expressed lower levels of *Hotair*

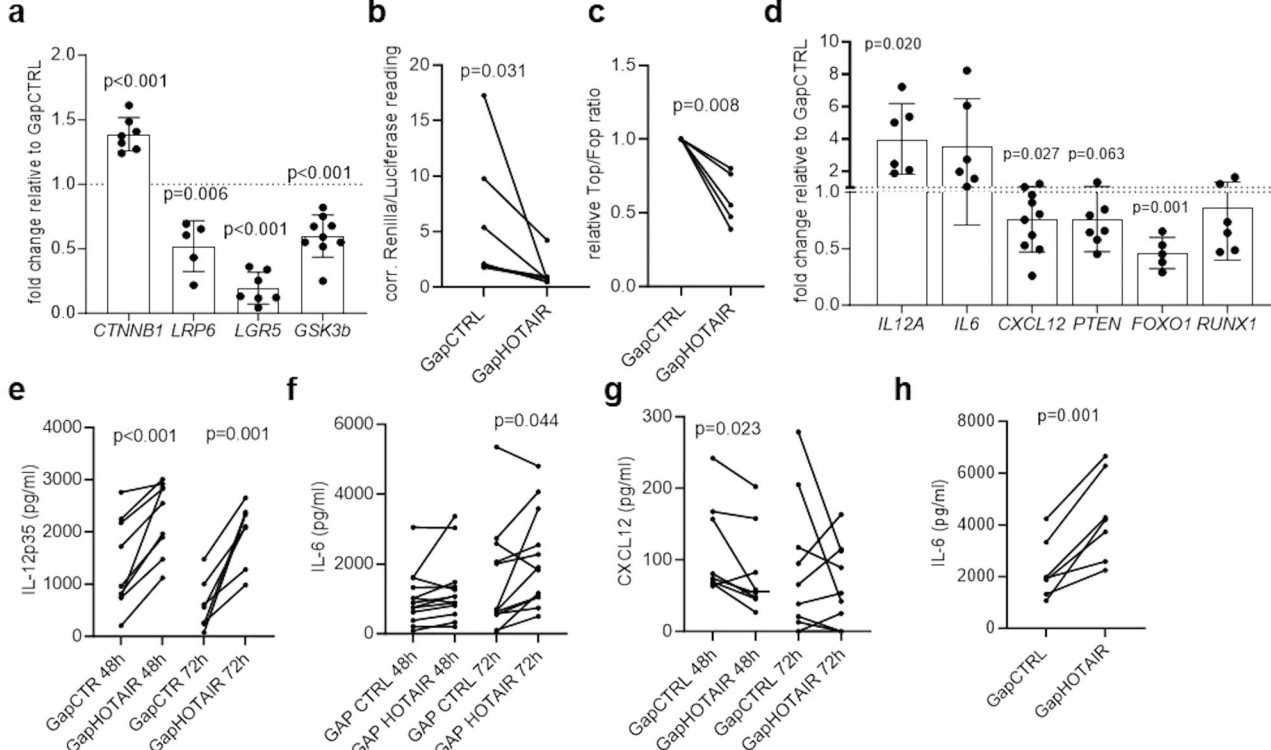

**Fig. 6 | *HOTAIR* modulates Wnt signaling and cytokine expression. a** Expression of *CTNNB1* (*n* = 8), *LRP6* (*n* = 5), *LGR5* (*n* = 7) and *GSK3B* (*n* = 9) measured by quantitative PCR between synovial fibroblasts (SF) transfected with control (GapCTRL) or *HOTAIR* targeting GapmeR (GapHOTAIR) after 48 h. Control transfected cells were set to 1, Mean +/− standard deviation is shown, one sample *t* test. **b** Activation of the canonical Wnt pathway was assessed by luciferase assay with a TCF/LEF reporter assay (*n* = 6) or (**c**) with a Wnt reporter gene (Top) or a mutated Wnt reported gene (Fop) as control (*n* = 5) in SF transfected with control or *HOTAIR* targeting GapmeR after 48 h.Wilcoxon and one sample *t* test, respectively.

**d** Expression of *IL12A* (*n* = 6), *IL6* (*n* = 6), *CXCL12* (*n* = 10), *PTEN* (*n* = 7), *FOXO1* (*n* = 7) and *RUNX1* (*n* = 6) measured by quantitative PCR between SF transfected with control or *HOTAIR* targeting GapmeR after 48 h. Control transfected cells were set to 1. Mean +/− standard deviation is shown. One sample *t* test. **e** IL12p35 (48 h: *n* = 9, 72 h: *n* = 7), (**f**) IL-6 (48 h: *n* = 13, 72 h: *n* = 12) and (**g**) CXCL12 (48 h: *n* = 8, 72 h: *n* = 9) were measured in supernatants of SF transfected with control or *HOTAIR* targeting GapmeR after 48 h and 72 h by ELISA. All paired *t* test. **h** IL-6 was measured in supernatants of micromasses after 7 days by ELISA (*n* = 7). Paired *t* test. Not reported *p*-values on the figure were not significant.

than SF from healthy wild-type mice (Fig. 4d), showing that the downregulation of *HOTAIR* in inflammatory conditions is conserved across species.

## *HOTAIR* regulates arthritis relevant pathways

We next examined the effect of downregulation of *HOTAIR* in inflammatory states in SF. Since *HOTAIR* represses gene expression by placing repressive H3K27me3 marks[21], we first analysed the effect of *HOTAIR* downregulation in SF on H3K27me3. A total of 2376 genomic sites with differential presence of H3K27me3 marks were identified between control SF and SF silenced for *HOTAIR* (silencing efficiency is shown in Supplementary Fig. S2a). The frequency of repressive H3K27me3 marks was decreased in promoters near transcription start sites in *HOTAIR*-silenced SF, showing a clear impact of *HOTAIR* downregulation on the epigenetic landscape of SF (Fig. 4e). We could confirm these data by using a CUT&Tag approach, which also showed decreased levels of H3K27me3 enrichment in SF after *HOTAIR* silencing (Supplementary Fig. S2b).

We then investigated the transcriptional changes associated with *HOTAIR* silencing in SF, mimicking an inflammatory arthritis environment. A total of 7885 genes were differentially expressed between control and *HOTAIR*-silenced SF (FDR < 0.05), with enrichment of site-specific signaling pathways such as MAPK, Wnt and PI-Akt signaling (Fig. 5a, Supplementary Data 3). RNA and ChIP sequencing suggested a *HOTAIR*-dependent downregulation of several collagen transcripts, including *COL1A1*, *COL1A2* and *COL3A1*, which could, however, not be confirmed by quantitative PCR (Fig. 5b). In contrast, there was a trend

towards increased procollagen release into supernatants from *HOTAIR*-silenced SF (48 h: *p* = 0.065 Wilcoxon matched-paired; 72 h: *p* = 0.053 paired *t* test) (Fig. 5c). Assuming that prolonged 3D culture systems provide a more natural environment for the production of extracellular matrix proteins from fibroblasts, we cultured control and *HOTAIR*-silenced SF in 3D micromass organ systems. After 3 weeks in culture, *HOTAIR*-silenced SF had deposited significantly more collagen-1 in micromasses compared to control SF (Fig. 5d). For further confirmatory analysis at the mRNA and protein levels, we focused on the most changed transcripts within the most relevant enriched pathways. Consistent with an increase in extracellular matrix remodelling in *HOTAIR*-silenced micromasses, several transcripts of the fibroblast growth factor (FGF) family, known to play a key role in extracellular matrix remodelling[32], were regulated by *HOTAIR*, albeit most of them were down-regulated. However, a particular strong upregulation of FGF8 after *HOTAIR* silencing was observed on mRNA (Fig. 5e) as well as on protein level (Fig. 5f). FGF were previously shown to signal via the PI-Akt and the Wnt signaling pathway and to play a crucial role in limb development[33,34]. Measurements of the expression of AKT and its active phophorylated form showed that *HOTAIR* silencing did not influence AKT levels, but resulted in increased AKT phosphorylation (Fig. 5g). Furthermore, we confirmed that silencing of *HOTAIR* regulated several transcripts in the Wnt signaling pathway (Fig. 6a) and repressed the activation of the canonical Wnt pathway in SF (Fig. 6b, c). Finally, we confirmed that *HOTAIR* regulated several cytokines and transcription factors which were previously implicated in SF activation in RA (*IL-12A*[35], *IL-6*[36], *CXCL12*[37], *PTEN*[38], *FOXO1*[39],

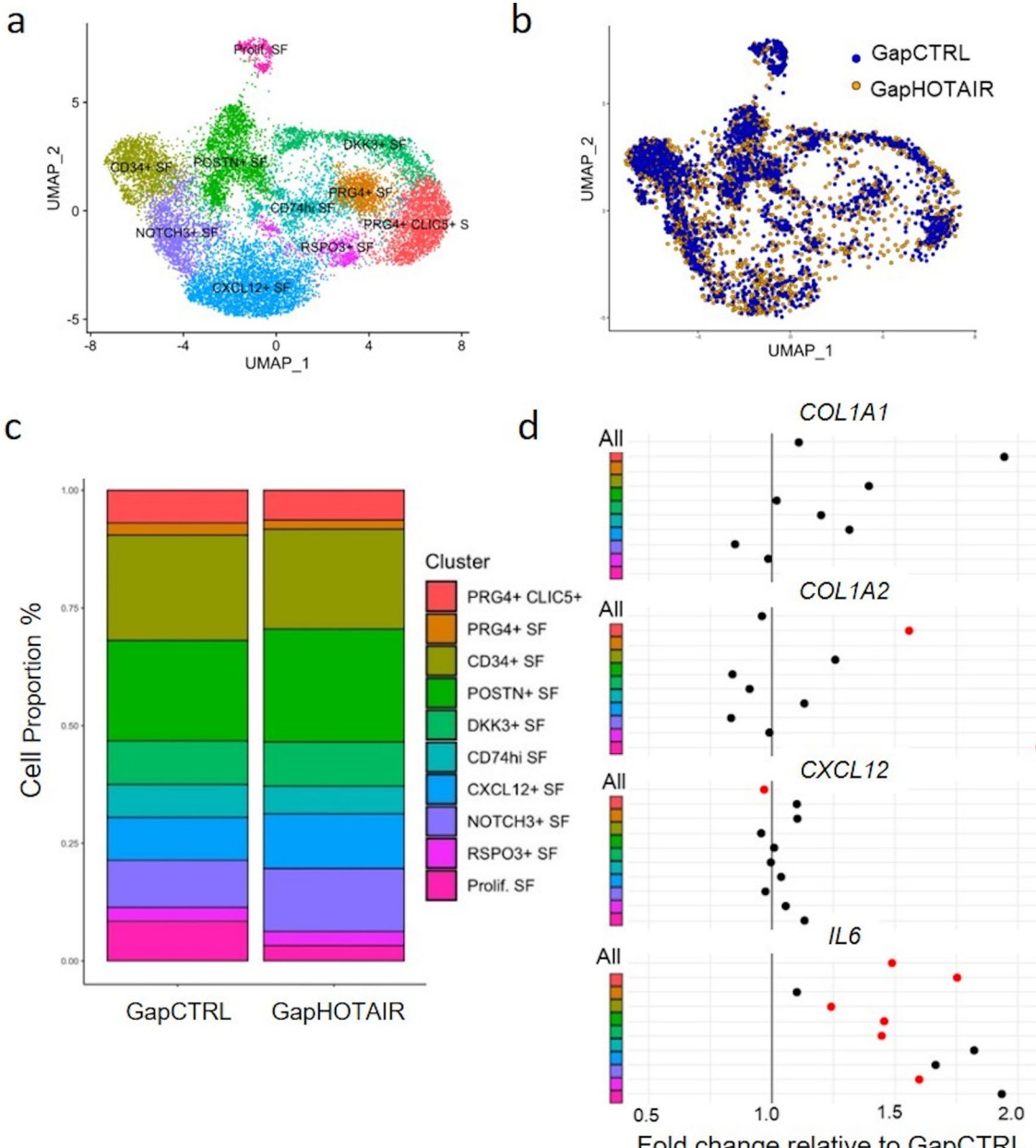

**Fig. 7 | Changes in *HOTAIR* expression modulate synovial fibroblast subtypes.**
**a** Single cell RNA sequencing (scRNAseq) data from synovial tissues were integrated with scRNAseq data from cultured synovial fibroblasts (SF) transfected with control (GapCTRL) or *HOTAIR* targeting GapmeR (GapHOTAIR) (*n* = 3). UMAP representation of the different SF subtypes is shown. **b** UMAP representation of the distribution of cells in control and *HOTAIR* silenced SF (cultured cells only). **c** Proportions of the different SF subtypes in control and *HOTAIR* silenced SF. x-squared = 80.947; df = 9; *p*-value = 10.48e-13. Chi-Square. **d** Change of expression of selected genes within the different SF subtypes. Red dots mark *p* < 0.05. Wilcoxon Rank Sum test.

*RUNX1*[40]) (Fig. 6d). IL-12p35 (Fig. 6e) and IL-6 (Fig. 6f) secretion by SF increased after the silencing of *HOTAIR*, while CXCL12 secretion decreased (Fig. 6g). Increased IL-6 levels were also released by micromasses formed with *HOTAIR* silenced SF compared to control SF (Fig. 6h). Most of these changes induced by *HOTAIR* silencing were already detectable 48 h after silencing (FGF8, Wnt inhibition, IL-12p35, CXCL12). However, phosphorylation of AKT and increased IL-6 secretion were only significant at later time points, suggesting indirect effects of decreased *HOTAIR* levels. Accordingly, inhibition of protein translation did not influence the effect of *HOTAIR* silencing on *CTNNB1*, *FGFR2* and *LGR5* (Supplementary Fig. S3a), but suppressed the effect of *HOTAIR* silencing on *GSK3B* and *FGF7* (Supplementary Fig. S3b), suggesting direct as well as indirect mechanims (e.g. mediated by an intermediate protein) of this regulation.Taken together, these data

clearly show that *HOTAIR* can regulate pathways relevant to joint inflammation and tissue remodelling in SF.

### Changes in *HOTAIR* expression modulate SF subtypes

Since several marker genes for recently described SF subpopulations[13,15,41,42] were affected by *HOTAIR* silencing, we wondered whether the observed transcriptional changes after *HOTAIR* silencing might be connected to changes in the proportion and the formation of SF subtypes. Thus, we integrated scRNAseq data from control and *HOTAIR*-silenced SF cultures with our previous meta-analysis of synovial tissue scRNAseq data from five different datasets[42]. We recapitulated nine key functional subpopulations of SF that have been described previously by scRNAseq analysis in synovial tissues (Fig. 7a, Supplementary Fig. S4a). CD90−/PRG4+ SF are considered as

lining SF, whereas CD90+ subpopulations are located in the sublining. The RSPO3+ SF population was described as intermediate lining/sublining phenotype[43]. In line with published data[14], several of the in vivo SF subtypes were partially lost during culture (Fig. 7b). The main SF subpopulations in culture were POSTN+ SF, CD34 + SF, NOTCH3+ cells and a mixed-marker cell population consisting of proliferating cells (prolif. SF) (Fig. 7a, b). Silencing of *HOTAIR* was associated with a small but significant change in the overall proportions of SF subpopulations (Fig. 7c). Individually, none of the changes in the SF subgroups reached statistical significance (Supplementary Figure S4b). However, there was a noticeable decrease in the proportion of proliferative SF and CD74hi SF in SF silenced for *HOTAIR*. POSTN+ SF and CXCL12+ SF were rather increased in the *HOTAIR*-silenced condition, even though *CXCL12* transcription itself was decreased in this subtype, which is the main producer of *CXCL12* (Fig. 7c and Supplementary Figure S4a, b). In contrast to *IL-6*, which was upregulated in all SF subtypes, but similar to *CXCL12, HOTAIR* regulated *COL1A1* and *COL1A2* expression different across SF subtypes (Fig. 6d). *COL1A1* and *COL1A2* were mainly increased in PRG4+ CLIC5+ SF after silencing, with no change or a decrease in the main collagen producing subtype, POSTN+ SF (Fig. 7d and Supplementary Fig. S4a). This might reflect a transcriptional switch from POSTN+ SF to collagen producing PRG4+ SF and might explain the inconsistent results seen in the bulk analysis of COL transcripts (Fig. 5b). From this analysis, it can be hypothesized that *HOTAIR* could play a role in the formation of SF subtypes, for example, by directing differential collagen or CXCL12 production between the SF subtypes. However, there were also regulatory mechanisms of *HOTAIR* that were evident in all SF subtypes.

### *HOTAIR* downregulation alters key functions in SF and surrounding cells

Next we aimed to decipher the functional changes of SF induced by *HOTAIR* downregulation upon inflammation. Real-time analysis of attachement and growth of SF in vitro did not show any changes in adhesion (Fig. 8a) and proliferation (Fig. 8c), but a decrease in spreading of *HOTAIR* silenced SF (Fig. 8b). Accordingly, silencing of *HOTAIR* resulted in decreased migration of SF (Fig. 8d and Supplementary Movie 1). These data are in line with the findings that *HOTAIR* regulated genes were enriched in the pathways "actin cytoskeleton" and "Wnt signaling" (Supplementary Data 3 and Fig. 5a), which were previously linked to tissue remodelling and cell migration[44,45]. Furthermore, *HOTAIR* silencing increased Fas-induced apoptosis in SF (Fig. 8e), as also indicated by the pathway 'apoptosis' in the pathway analysis (Supplementary Data 3).

Since osteoclast differentiation is of major importance in RA and was differentially expressed between hand and knee SF (Supplementary Data 2), we assessed the effect of silencing *HOTAIR* in SF on osteoclastogenesis and osteoclast function. Co-culture of differentiating monocytes with *HOTAIR* silenced SF, but not the addition of supernatants from *HOTAIR* silenced SF, decreased osteoclast formation (Fig. 8f), suggesting that cell-cell contact was needed to inhibit osteoclastogenesis. In contrast, osteoclasts in co-culture with *HOTAIR* silenced SF, as well as incubated with supernatants showed decreased osteoclast activity (Fig. 8g).

Since the levels of several chemokines and cytokines were affected by *HOTAIR* silencing (Supplementary Data 3), we compared the chemotactic activity of supernatants derived from control or *HOTAIR* silenced SF. Despite a similar amount of healthy peripheral blood mononuclear cells (PBMCs) migrating towards conditioned supernatants from controls and *HOTAIR* silenced SF (Fig. 9a), we observed a shift in the cellular composition of the migrated PBMCs with an increased number of CD19 + B cells (Fig. 9b) and a decreased number of CD14+ monocytes (Fig. 9c) using supernatants from *HOTAIR* silenced SF. A slight increase was also seen for the chemotaxis of CD3 + T cells (Fig. 9d). In line with these results, synovia of RA and OA

patients with low *HOTAIR* expression were characterized by higher CD20 + B cell (Fig. 9e) and, in particular, CD138+ plasma cell infiltration (Fig. 9f) and a lymphoid histological pattern (Fig. 9g: RA synovial tissues and Supplementary Fig. S5: OA synovial tissues). Consistently, in the early RA cohort (PEAC)[46], low synovial *HOTAIR* expression in synovium was associated with a trend towards a lymphoid histological pattern (Fig. 9h) and *HOTAIR* expression was negatively correlated with CD138+ plasma cell infiltrates (r: 0.27, padj: 0.023) (Fig. 9i). Although the correlation was weak, this result was supported by the previous findings in Fig. 8f–h. In summary, these data support the notion that expression levels of *HOTAIR* in SF can shape the influx of immune cells in the synovium in arthritis and vice-versa.

### *HOTAIR* site-specific expression may shape cellular response in other organs

Since HOX gene expression is site specifically expressed in stromal cells of several organs and tissues, we wondered whether *HOTAIR* might shape the inflammatory response in other tissues than joints. Site-specific expression of *HOTAIR* was already shown in human skin where it also follows the upper vs lower body part pattern[20]. In addition, we measured site-specific expression of *Hotair* in mouse spine and found increased expression in lumbal compared to cervical spine ($p = 0.009$) (Fig. 9j). Furthermore, in the different anatomic compartments of the gastrointestinal tract, *Hotair* showed site-specific expression with higher expression in the distal parts of the intestines compared to stomach and the small intestines (Fig. 9k).

### Discussion

Here, we show that *HOTAIR* is a major regulator of site-specific gene expression in SF and modulates a series of highly relevant signalling pathways and SF functions in arthritis. *HOTAIR*-modulated changes in SF gene expression and function were associated with changes betweeen hand and knee arthritis in RA, suggesting that *HOTAIR* may shape the phenotype of arthritis in lower extremity joints. By showing that an embryonically imprinted, site-specific factor can regulate inflammation-related signalling pathways, our data support the concept that anatomically defined features of the local stroma can influence the susceptibility and manifestation of inflammation.

Here, we showed that *HOTAIR* expression in SF is decreased after stimulation by local inflammatory factors, which could explain its decreased expression in RA SF. Other recent studies have shown that *HOTAIR* expression could be regulated by various factors in the local microenvironment, such as hypoxia[47], hormons[29] or inflammatory factors[48]. The ability of external factors to influence *HOTAIR* expression supports the idea that embryonic site-specific expression of *HOTAIR* is used to trigger a locally distinct and anatomically restricted stress response. Interestingly, it has been suggested that *HOTAIR* may also be mechanoresponsive in response to stretch[49]. Given the increased expression of *HOTAIR* in the lower limbs and lumbar spine, load and mechanosensing could be additional factors that regulate site-specific response pathways via *HOTAIR*.

In our study, inflammation-induced downregulation of *HOTAIR* modulated several inflammatory response pathways in SF, such as the MAPK, PI-Akt and canonical Wnt pathways. Consistent with our results, *HOTAIR* silencing inhibited the canonical Wnt pathway in gastric and pancreatic cancer cells[50,51] and in OA chondrocytes[52,53]. In OA chondrocytes[53], *HOTAIR* has been shown to act directly on Wnt inhibitory factor 1 (WIF-1) by increasing histone H3K27me3 in the *WIF-1* promoter, leading to WIF-1 repression that promotes activation of the Wnt/β-catenin pathway. Consistent with this study, our own results based on the adjunction of cycloheximide suggest an indirect mechanism underlying the *HOTAIR*-mediated regulation of the Wnt pathway.

Activation of the Wnt pathway is a characteristic of the pauci-immune subtype of synovitis in RA, whereas more inflammatory

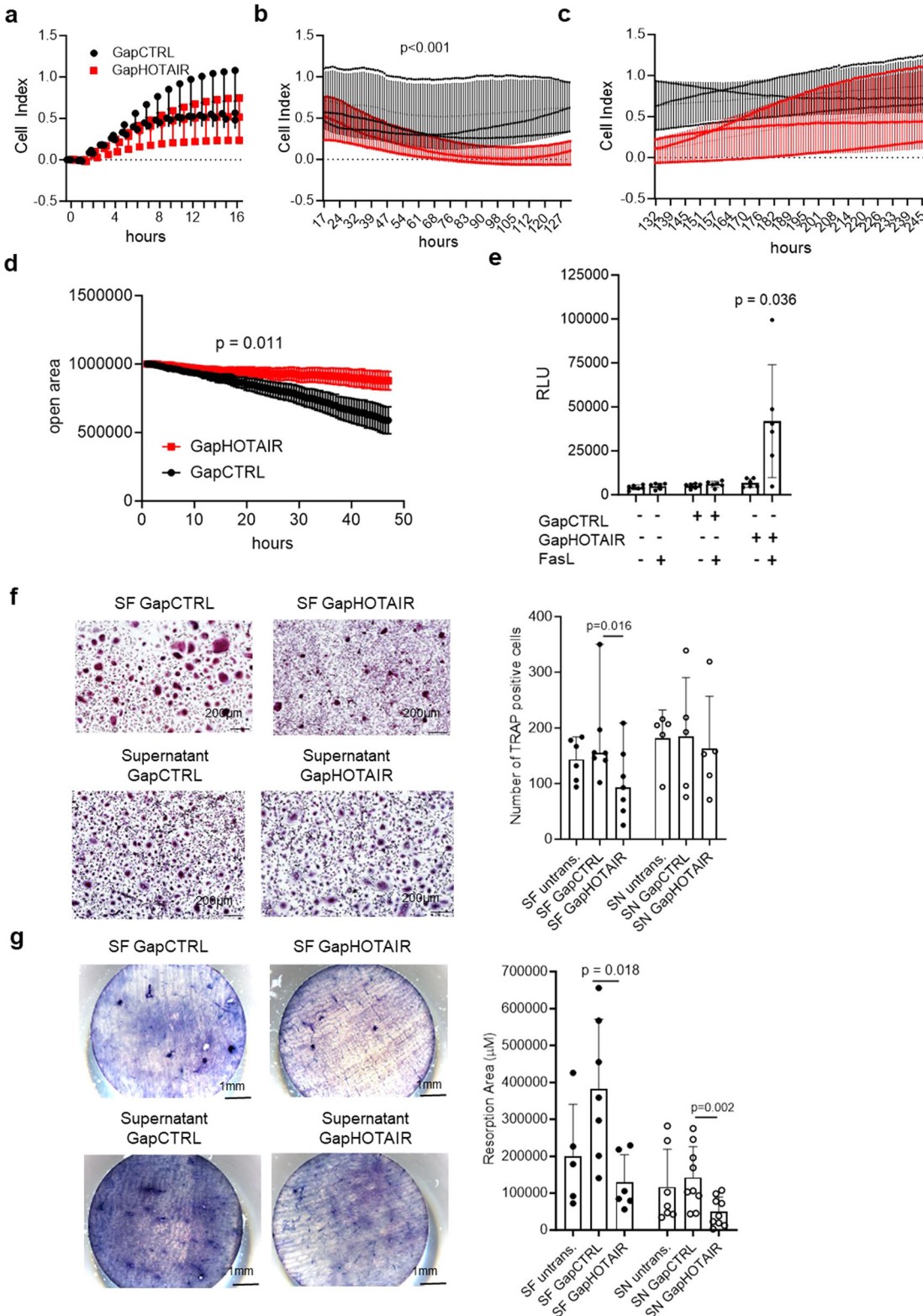

pathways such as PI-Akt have been found to be activated in lymphoid/myeloid synovial histological patterns[6,8,9]. Consistently, lower levels of *HOTAIR* in synovial tissue were associated with a lymphoid histological pattern in our study. Furthermore, *HOTAIR* silencing exerted a chemotactic effect on lymphocytes in vitro. Thus, it can be speculated that *HOTAIR* acts as a stromal regulator of the inflammatory tissue response, whose downregulation under conditions with high levels of

TNF might promote the development of a lymphocyte-dominated inflammatory response.

Studies analysing the differences between RA in different joints are rare. Our results suggest that hand RA is more likely than knee RA to show the classic signs of synovitis described for RA, whereas the pauci-immune histological pattern of synovitis was more common in knee RA than in hand RA. Refractory, difficult-to-treat arthritis has

**Fig. 8 | _HOTAIR_ silencing induces functional changes in synovial fibroblasts.**
Real-time measurements of synovial fibroblasts (SF) (**a**) adhesion (0–16 h), (**b**) spreading (17–127 h) and (**c**) proliferation (132–245 h) in SF transfected with control GapmeR (GapCTRL) or _HOTAIR_ GapmeR (GapHOTAIR) ($n = 3$). Cell index = arbitrary unit of impedance. Mean +/− standard deviation is shown. Two-way ANOVA. **d** Measurement of open area covered over time by SF transfected with control GapmeR or _HOTAIR_ GapmeR ($n = 6$) in scratch assay. Mean +/− standard deviation is shown. Two-way ANOVA. **e** Caspase 3/7 activation in untransfected, control and _HOTAIR_ GapmeR transfected SF after 48 h of transfection ($n = 6$). RLU relative luminescence units after background subtraction. Mean +/− standard deviation is shown. Paired _t_ test. **f** Left panel: representative pictures of tartrate-resistant acid phosphatase (TRAP) staining of osteoclasts differentiated from monocytes by co-culture ($n = 7$) (upper panel) or incubation with supernatants (SN) ($n = 5$) of control or _HOTAIR_ transfected SF (lower panel). Right panel: quantification of TRAP+ cells in the described conditions. SF: co-culture with synovial fibroblasts (black dots), median and range is shown. Wilcoxon paired test; SN: incubation with supernatant (white dots). Mean +/− standard deviation is shown. **g** Left panel: representative pictures of resorption areas after incubation of bone slices with osteoclasts differentiated by co-culture ($n = 6$)(upper panel) or incubation with supernatants of control or _HOTAIR_ transfected SF ($n = 9$) (lower panel). Right panel: quantification of resorption areas in the described conditions. SF: co-culture with synovial fibroblasts (black dots); SN: incubation with supernatant (white dots). Mean +/− standard deviation is shown. Paired _t_ test. Not reported _p_-values on the figure were not significant.

been associated with this non-inflammatory pauci-immune histological pattern[10], however, up to now it is not known whether lower limb synovitis is more often resistant to immunosuppressive therapy. In depth analysis of the characteristics of knee RA has suggested severe cartilage destruction but less bone erosion compared to what is known from RA of the hand[54]. Similarly, OA of the hand tends to be more erosive than OA of the knee[55]. The less destructive phenotype of RA and OA in knees may be due to protective mechanisms exerted by the joint-specific stroma. Both, our comparison of knee and hand RA as well as our analysis of _HOTAIR_ function pointed towards decreased osteoclastogenesis in knees compared to hands. Co-culture of differentiated monocytes with _HOTAIR_-silenced SF, but not the addition of _HOTAIR_-silenced SF supernatants, decreased osteoclast formation, suggesting that cell-cell contact was necessary to inhibit osteoclastogenesis. Thus, the decrease in osteoclastogenesis after _HOTAIR_ silencing could be related to the reduced migratory function of SF silenced for _HOTAIR_[48]. Another explanation could be that the decrease in osteoclastogenesis is mediated by the decreased CXCL12 secretion following _HOTAIR_ silencing[56], as osteoclasts co-cultured with _HOTAIR_ silenced SF but also incubated with supernatants showed decreased osteoclast activity[57].

One limitation of our study is that with the available data we cannot directly show how down-regulation of _HOTAIR_ during inflammation affects the inflammatory response in a joint-specific manner. In vivo confirmation of our hypotheses is hampered by the fact that _Hotair_ does not have the same function in morphological development in mice and humans[58], and it is therefore questionable whether the function in regulating inflammatory pathways is conserved. These differences between the species might be explained by biomechanical and anatomical differences between quadrupedal walking in mice versus bipedal walking in humans[49,59]. If our hypothesis of an influence of site-specific embryonic factors on inflammatory responses is valid, species-specific differences in limb formation, anatomy and biomechanics may underlie the limited translation of preclinical studies in mice to human arthritis. Another limitation is, that by using GapmeR technology, we did not obtain complete and long-lasting silencing of _HOTAIR_ as could be achieved with CRISPR-based technology, for example. However, as our aim was to mimic the situation in vivo in a pro-inflammatory environment, i.e. a decrease in _HOTAIR_ expression without complete silencing, the use of Gap_HOTAIR_ appeared to be the most suitable approach. However, we recognize that even with this approach, we are not able to fully recapitulate what happens in an inflammatory environment. Moreover, we acknowledge that _HOTAIR_ is not the only regulator of the differential transcriptome between hands and knees. Other HOX genes are differentially expressed between joints as highlighted in our Fig. 1e, g. Their effect on the transcriptome and fibroblast function will need to be investigated in further studies. Furthermore, different mechanical and anatomical influences between joints might have a great impact on joint specific gene expression.

Beyond the joint, _HOTAIR_ appears to be involved in the site-specific regulation of inflammation in other organs. Thus, it has recently been identified as a major regulator of region-specific development of adipose tissue, which is associated with site-specific metabolic complications, with exclusive expression in gluteofemoral subcutaneous adipose tissue[29,60]. In addition, _HOTAIR_ is mostly expressed in distal dermal fibroblasts[20]. Consistently, we observed site-specific expression of _HOTAIR_ in the spine and intestine with increased expression in the lumbar spine and distal parts of the intestines. In spondyloarthritis, it has been suggested that involvement of the lumbar spine is more frequent[61] and more severe with more bone bridges compared to cervical involvement[62]. Furthermore, Wnt signaling plays a crucial role in the formation of bone bridges in spondyloarthritis[63]. Similarly, inflammatory bowel disease encompasses two types of idiopathic intestinal disease that are differentiated by their location and the depth of involvement in the bowel wall[64]. Ulcerative colitis most commonly affects the rectum, whereas Crohn's disease most often affects the terminal ileum and colon. Interestingly, Wnt signaling has been identified as a key regulatory pathway in the intestinal mucosa[65]. Thus, important future work will be to elucidate whether the site-specific expression of _HOTAIR_ underlies the development of site-specific phenotypes of these inflammatory diseases.

In conclusion, we suggest that the phenotype and severity of inflammation are modulated by the embryonic imprinted local stromal gene signature. Further investigation into joint specific factors favouring and shaping the development of arthritis will improve understanding of the pathogenesis of arthritis and could lead to the development of specific therapies targeting joint specific signaling pathways.

## Methods
### Patients
Synovial tissues were obtained from OA and RA patients undergoing joint replacement surgery at the Schulthess Clinic Zurich, Switzerland or from ultra-sound guided joint biopsies. RA patients fulfilled the 2010 ACR/EULAR (American College of Rheumatology/European League Against Rheumatism) criteria for the classification of RA[66] whereas OA was considered in cases of chronic pain and manifest radiographic signs of OA[67] without any underlining inflammatory rheumatic disease. Patients' characteristics are given in Supplementary Table S2.

### Culture of SF
**Human SF**. Synovial tissues were digested with dispase (37 °C, 1 h) and SF were cultured in Dulbecco's modified Eagle's medium (DMEM; Life Technologies) supplemented with 10% fetal calf serum (FCS), 50 U ml$^{-1}$ penicillin/streptomycin, 2 mM L-glutamine, 10 mM HEPES and 0.2% amphotericin B (all from Life Technologies). For functional assays, vitamin C (50 μg/ml) was added to the culture medium. Purity of SF cultures was previously confirmed by flow cytometry showing the presence of the fibroblast surface marker CD90 (Thy-1)(555595, BD Bioscience, 1ul, 1:200) and the absence of leukocytes (CD45)(130-098-043, Miltenyi, 1ul, 1:200), macrophages (CD14) (Clone REA599, 130-110-519, Miltenyi, 1 μl, 1:200), T lymphocytes (CD3) (clone UCHT1, 11-0038-42, Invitrogen, 1 μl, 1:200), B lymphocytes (CD19) (Clone HIB19,

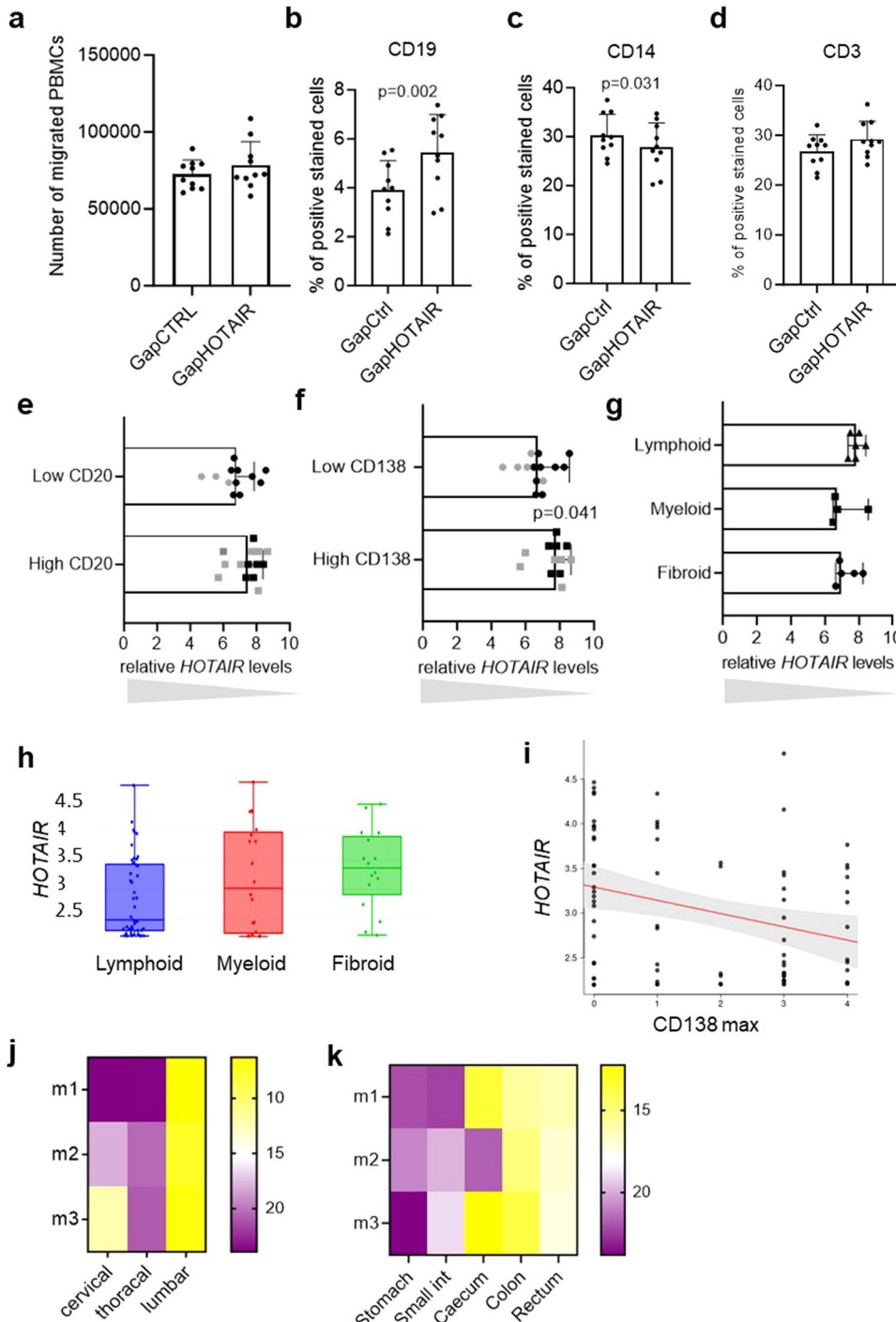

**Fig. 9 | *HOTAIR* silencing increases lymphocyte chemotaxis. a** The amounts of PBMCs migrating through a transwell system towards conditioned supernatants from synovial fibroblasts (SF) transfected with control GapmeR (GapCTRL) or *HOTAIR* GapmeR (GapHOTAIR) (*n* = 10). Mean +/− standard deviation is shown. **b–d** The percentage of (**b**) CD19, (**c**) CD14 and (**d**) CD3 positive cells was measured in PBMCs that had migrated towards supernatants of control or *HOTAIR* silenced SF (*n* = 10). Mean +/− standard deviation is shown. Paired *t* test. **e–g** Expression levels of *HOTAIR* were measured by qPCR in synovial tissues from rheumatoid arthritis (RA) (black dots) and osteoarthritis (OA) (grey dots) with (**e**) high (CD20+ score ≥2) (*n* = 14) or low amounts of CD20 + B cells (*n* = 14) (mean +/− standard deviation is shown), (**f**) with high (CD138+ score ≥ 2) (*n* = 12) or low amounts of CD138+ plasma cells (*n* = 14) (median and range is shown, Mann–Whitney test), (**g**) with pauci-immune/fibroid (*n* = 5), myeloid (*n* = 4) or lymphoid (*n* = 6) histological patterns in RA (median and range is shown). **h**, **i** Expression levels of *HOTAIR* in RNA sequencing in synovial tissues from the PEAC cohort according to (**h**) pauci-immune/fibroid (*n* = 16), myeloid (*n* = 18) or lymphoid (*n* = 43) histological patterns (min max and median are shown) and to (**i**) CD138 infiltrates. **j**, **k** Expression of *Hotair* was measured by qPCR in C57/BL6 mice (*n* = 3) (**j**) in cervical, thoracal and lumbal spine and in (**k**) different parts of the gut (stomach, small intestine, caecum, colon and rectum). Heatmap of the relative *HOTAIR* expression according to the localisation are presented. Not reported p-values on the figure were not significant.

Ref 12–0199, Invitrogen, 1 μl, 1:200) and endothelial cells (CD31) (555445, BD Biosystems, 1 μl, 1:200)[17]. Cell cultures were negative for mycoplasma contamination as assessed by MycoAlert mycoplasma detection kit (Lonza). SF from passages 5 to 8 were used. The experiments were performed with different samples.

**Murine SF.** Wild-type and TNF transgenic (hTNFTg) C57BL/6 mice (n = 14 and 13) were generated as previously described[68] and housed under specific pathogen-free conditions in the animal facilities of the Biomedical Sciences Research Centre (BSRC) Alexander Fleming. All experiments were approved by the local ethics committee and in accordance with the institutional care and use committee of BSRC Alexander Fleming. Fully diseased hTNFTg and healthy wild-type mice were killed at 6–8 weeks of age. The sample size for animal experiments was chosen based on previous laboratory experience and published evidence from other groups. No randomization was used and no blinding was done. hTNFTg animals have a 100% penetrance of joint disease. At the age of 6–8 weeks, joint disease is considered as established. Mice were randomly selected from three individual breedings (hTNFTg and wild-type controls) and each group was co-housed. hTNFTg and wild-type littermate controls were randomly chosen from the litters. Primary mouse SF were isolated from ankles (n = 15, 7 from hTNFTg mice and 8 from wild-type controls) and knees (6 from each) and cultured for four passages, as previously described[69]. The purity of mouse SFs was determined as follows: >80% CD90.2+ (05316, Biolegend, 1:300) and <5% CD11b+ (557397, BD Bioscience, 1:400), with >90% Vcam1+ (105712, Biolegend, 1:400) within CD90.2+ population as previously described[70].

## Histological analysis

Formalin-fixed, paraffin-embedded synovial tissues of RA or OA patients were cut, put on slides and stained with hematoxylin/eosin. 36 synovial tissues from hands (22 from joint replacement and 14 from synovial biopsies) and 27 from knee (18 from joint replacement and 9 from synovial biopsies) were assessed. Synovitis score was assessed by evaluation of the thickness of the lining cell layer, the cellular density of synovial stroma and leukocyte infiltration as described by Krenn et al.[30]. Vascularization was assessed by counting the amount of CD31+ cells (ab28364, Abcam; dilution: 1:50) on 5 consecutive pictures on 20× objective. Synovial tissue were stained with CD3 (ab16669, Abcam; dilution: 1:200), CD20 (M0755, Agilent/Dako; dilution: 1:50), CD138 (Clone MI15, Agilent/Dako; dilution: 1:50) and CD68 (M0814, Agilent/Dako; dilution: 1:50), to stratify them into lymphoid, myeloid and fibroid histological patterns according to previously published histological features[7,9,30].

## Gene silencing

Hand SF were transfected with 50 nM antisense LNA *HOXD10* (Qiagen, Sequence: 5′-TGT CTG CGC TAG GTG G-3′), *HOXD11* (Qiagen, Sequence: 5′-TGC TAG CGA AGT CAG A-3′), *HOXD13* (Qiagen, Sequence: 5′-CAT CAG GAG ACA GTA T-3′) or *HOTTIP* (*HOTTIP*1 Qiagen, Sequence: 5′-TCG GAA AAG TAA GAG T-3′ and *HOTTIP*2 Qiagen, Sequence: 5′-TAC CTA AGT GTG CGA A-3′) GapmeR. Knee SF were transfected with 50 nM antisense LNA *HOTAIR* GapmeR (Qiagen, Sequence: 5′-AGG CTT CTA AAT CCG T-3′), CBP (Qiagen, Sequence: 5′-GCG GCG ATC CTT TAG A-3′) or p300 (Qiagen, Sequence: 5′-TAG TCT GGT CCT TCG T-3′). Transfections were performed using Lipofectamine 2000 (Invitrogen) according to the manufacturer's instructions. Antisense LNA GapmeR Negative Control A (Cat No 300610) was used as transfection control. For the intersectional analysis of differentially expressed genes between hand and knee (adjusted p-value < 0.05) regulated by the HOX family, we used the R package UpSetR v14 und ComplexUpset v1.3.3[71,72].

## Single-cell RNA sequencing

**scRNAseq of hand and knee synovial tissues.** ScRNAseq sequencing was performed on ultrasound-guided joint biopsies of hand (n = 8) or knee (n = 6) joints of RA patients (Patient's characteristics in Supplementary Table S1). Tissues were processed as previously described[42]. In brief, tissues were mechanically minced and enzymatically digested using Liberase TL (Roche). 10'000 unsorted synovial cells (viability >80%) per patient were prepared for single cell analysis using the Chromium Single Cell 3′ GEM, Library and Gel Bead Kit v3.1, the Chromium Chip G Single Cell Kit and the Chromium controller (all 10× Genomic). Libraries were sequenced on the Illumina NovaSeq6000 instrument to a sequence depth of 20,000 to 70,000 reads per cell. CellRanger (v2.0.2) from 10× Genomics was used to demultiplex, align the reads to Ensembl reference build GRCh38.p13 and collapse unique molecular identifiers (UMIs). The standard Seurat protocol[73] was used for further analysis. Gene expression between hand and knee was compared. Pathway enrichment analyses of genes differentially expressed between SF from hands and knees were performed using Enrichr (all genes with FDR < 0.05, log fold change +/−1)[74]. The scatter dot plots were created with the Enrichr Appyter.

**scRNAseq of control SF and SF silenced for HOTAIR.** scRNAseq in cultured SF transfected with either control or *HOTAIR* targeting GapmeR (n = 3) (Patient's characteristics in Supplementary Table S3) was performed. SF were washed and counted on a LUNA automated cell counter (Logos Biosystems). 15'000 unsorted SF (viability 60–88%) per patient were prepared for single cell analysis using the Chromium Single Cell 3′ GEM, Library and Gel Bead Kit v3.1, the Chromium Chip G Single Cell Kit and the Chromium controller (all 10× Genomic). Libraries were sequenced on the Illumina NovaSeq6000 instrument to a sequence depth of 20,000 to 70,000 reads per cell. CellRanger (v2.0.2) from 10× Genomics was used to demultiplex, align the reads to Ensembl reference build GRCh38.p13 and collapse unique molecular identifiers (UMIs). scRNAseq for control SF and SF silenced for *HOTAIR* were integrated with SF of publicly available datasets[13,15,41,75] and two in-house datasets[42]. The standard Seurat (version 4.0) protocol[73] was followed for analysis. Quality control included the exclusion of cells with >25% mitochondrial reads and <5% ribosomal reads; exclusion of mitochondrial, ribosomal and hemoglobin genes and inclusion of only cells with at least 200 detected genes. For the integration of all datasets, we applied harmony[76] within the standard Seurat workflow and performed the analysis as previously described[42]. Cell clustering was computed with 30 principal components and resolution of 0.1. Marker genes were identified by Wilcoxon Rank Sum test and log fold change of 0.25. Differential gene expression analysis for selected genes (*COL1A1, COL1A2, CXCL12* and *IL6*) between control and *HOTAIR* targeting GapmeR transfected SF were performed within each SF cluster applying Wilcoxon Rank Sum test and FDR p-value adjustment for multiple testing.

## In situ hybridization (ISH)

**Construction of probes.** PCR on cDNA generated from total RNA of knee SF was used to produce an amplicon of 267 base pairs of *HOTAIR* (see Supplementary Table S4). Amplicons were cloned into pPCR-Script Amp SK (+) plasmids using the PCR-Script Amp cloning kit (Agilent Technologies). Plasmids were amplified and purified with the PureLink MiniPrep kit (Thermo Fisher Scientific). Plasmids were linearized using restriction enzymes (EcoRI or NotI; both New England Biolabs) and purified with the QIAquick PCR Purification Kit (Qiagen). DIG-labeled *HOTAIR* probes were prepared by in vitro transcription with RNA polymerases and plasmid vectors containing target transcript sequences. Linearized plasmid DNA (10 μl) was used as a template, and RNA probes were synthesized with T7 or T3 RNA polymerase (Roche) and DIG Labeling Mix (Roche) for 100 min at 37 °C.

**In-situ hybridization.** *HOTAIR* expression was examined by ISH in paraffin-embedded synovial tissue from OA and RA patients (for patients' characteristics see Supplementary Table S2). All steps prior to and during hybridization were conducted under RNase-free conditions. Sections were deparaffinized, and incubated with HCl (20 min) and PFA 4% (10 min). Then sections were treated with trypsin (1 mg/ml) at 37 °C for 30 min. The slides were incubated with 2X SSC 5 min and washed twice with triethanolamine-HCl solution. The sections were acetylated for 20 min with 0.25% acetic anhydride in 0.1 M triethanolamine (pH 8.0) and washed twice in triethanolamine-HCl. Following incubation with hybridization buffer for 1 h at RT, the sections were incubated with hybridization solution which contained 1:10 diluted DIG-labelled probes in hybridization buffer (50% deionized formamide, 40% dextran sulfate/SSC solution, 50× Denhardt's solution, 5% pre-heated herring sperm DNA and 250 µg/ml tRNA). Slides were then incubated in a humidified chamber at 50 °C overnight. After hybridization, slides were washed with 5X SSC at 50 °C (20 min), with 50% formamide in 2X SSC at 50 °C (30 min) and two times with STE buffer (500 mM NaCl, 1 mM EDTA, 20 mM TRIS-HCL pH 7.5). Sections were treated with RNase A (40 µg/ml) in STE buffer at 37 °C for 1 h, followed by successive washing with STE buffer (RT), 2X SSC (RT), buffer 1 (0.2% SDS in 1X SSC, 50 °C), buffer 2 (0.2% SDS in 0.5X SSC; 50 °C) and buffer 3 (0.2% SDS in 0.1X SSC; 50 °C). Slides were incubated with 2% horse serum at RT for 30 min. Then the slides were incubated in a humidified chamber at RT for 1 h with sheep anti-digoxigenin-AP-Fab (11093274910, Roche), diluted to 1:250 in TBS-T containing 1% blocking reagent. Slides were washed with TBS-T and stained with nitro blue tetrazolium/5-bromo-4-chloro-3-indolylphosphate (Roche) in the dark. The intensity of staining was quantified with ImageJ software (http://rsbweb.nih.gov/ij/docs/examples/stained-sections/index.html). Negative controls were conducted by the substitution of sense for anti-sense probes or by the omission of anti-sense probes in the hybridization solution.

### Immunohistochemistry

For double staining of ISH slides, tissue sections were pre-treated with proteinase K 10 min at 37 °C. Endogenous peroxidase activity was disrupted with 3% $H_2O_2$. Slides were permeabilized with 0.1% Triton in PBS. Nonspecific protein binding was blocked with 10% goat serum in antibody diluent (DakoCytomation) for 1 h. Mouse anti-human CD68 (M0814, Agilent/Dako, Lot 20072689, Dilution 1:50) antibodies, mouse anti-human vimentin (ab7752, Abcam, Lot 892055, Dilution 1:100) antibodies or mouse IgG1 were applied over night at 4 °C. Slides were washed in PBS-T (0.05% Tween 20 in PBS) and incubated with biotinylated goat anti-mouse antibodies (115-066-068, Jackson ImmunoResearch; dilution: 1:1000). The signal was amplified with ABC reagent and detected with AEC (Vector laboratories).

Staining was imaged on a Zeiss Imager.Z1 (25× magnification) and quantified with ImageJ using brightness values.

### Quantitative real-time polymerase chain reaction (qPCR)

Snap frozen synovial tissues were minced and total RNA was isolated using the miRNeasy Mini kit (Qiagen) including on-column DNaseI digestion. From cultured cells, total RNA was isolated using the Quick-RNA MicroPrep Kit (Zymo) including on-column DNaseI digestion.

Total RNA was reversed transcribed and qPCR was performed using SYBR green (Life Technologies) or TaqMan probes for the detection of *HOTAIR*. Primer sequences are available in Supplementary Table S4. Changes after cycloheximide adjunction (10 ug/ml 6 h and 24 h) were evaluated (*n* = 3). No template control samples, dissociation curves and samples containing the untranscribed RNA were measured in parallel as controls. Data were analyzed with the comparative $C_T$ method and presented as $2^{-\Delta CT}$ or $2^{-\Delta\Delta CT}$ as described[77]. Constitutively expressed *HPRT* was measured for internal standard sample normalization in humans and *beta2-microglobulin* in mouse

SF. *COL3A1* was measured to normalize *HOTAIR* expression to the fibroblast content in synovial tissues and to ensure that differences were not due to variable numbers of fibroblasts in the synovial tissues.

### SF stimulation

SF were stimulated with human recombinant TNF (10 ng/ml; R&D Systems), human recombinant IL1β (1 ng/ml; R&D Systems), lipopolysaccharide (LPS) from *Escherichia coli* J5 (100 ng/ml; List Biological Laboratories), polyI-C (PIC) (10 µg/ml; InvivoGen), bacterial lipopeptide (bLP) palmitoyl-3-cysteine-serine-lysine-4 (300 ng/ml; InvivoGen) or human recombinant TGFβ (10 ng/ml; R&D Systems) for 24 h. For DNA demethylation, SF were treated daily with complete DMEM 10% FCS with 1 µM 5-azacytidine (Sigma Aldrich) for 5 days.

### Cap analysis gene expression (CAGE)

CAGE data from control or TNF stimulated (10 ng/ml; 24 h) RA SF from 2 knees were obtained from GSE163548 (GEO repository). Mapping and identification of CAGE transcription start sites (CTSSs) were performed by DNAFORM (Yokohama, Kanagawa, Japan). In brief, the sequenced CAGE tags were mapped to hg19 using BWA software and HISAT2 after discarding ribosomal RNAs. Identification of CTSSs was performed with the Bioconductor package CAGEr (version 1.16.0)[78]. Promoter and enhancer candidate identification and quantification were performed with the Bioconductor package CAGEfightR[79] (version 1.6.0) with default settings. Clusters were kept when present in at least one sample.

### ChIP DNA sequencing

SF pellets from OA knees transfected with GapmeR HOTAIR and GapmeR Control (*n* = 3 each) were prepared using the iDeal ChIP seq kit for Histones (Diagenode) with a shearing of 12 cycles (30''ON 30'' OFF, Bioruptor Pico). The shearing efficiency was analyzed using an automated capillary electrophoresis system Fragment Analyser (High sensitivity NGS fragment kit) after RNase treatment, reversion of crosslinking and purification of DNA. ChIP assays were performed using 1 million cells per IP and H3K27me3 (1 µg, C15410195, Diagenode). A control library was processed in parallel using the same amount of control Diagenode ChIP'd DNA. After the IP, the ChIP'd DNA was analyzed by qPCR to evaluate the specificity of the reaction. The promoter of GAPDH (GAPDH-TSS) was used as negative control region, Myelin Transcription Factor 1 gene (MYT1) was used as a positive control region. The ratios of the recovery for the positive regions over the background, i.e. the specificity of the signal, were substantially smaller in two of the samples (one GapmeR HOTAIR and one GapmeR control) (ratio MYT1/GAPDH-TSS 74 and 25, respectively compared to a mean of 122 ± 33 in the other samples). These samples also did not cluster with the other samples in unsupervised principal component analysis. Therefore, these 2 samples were excluded for the analysis. Libraries were prepared from 1 ng of IP and input DNA using the MicroPLEX v2 protocol, quantified by BioAnalyzer, purified (AMPure beads) and eluted in TE. Purified libraries were quantified (Qubit ds DNA HS kit), analysed for size (Fragment Analyzer) and diluted to 20 nM concentration. Libraries were sequenced on an Illumina HiSeq 2500 (50 bp, single end).

The quality of sequencing reads was assessed using FastQC. Reads were aligned to the reference genome (hg19) using BWA v. 0.7.5a[80]. Samples were filtered for regions blacklisted by the ENCODE project[81,82]. Subsequently samples were deduplicated using SAMtools version 1.3.1[83]. Alignment coordinates were converted to BED format using BEDTools v.2.17[83]. Peaks were annotated on gene and transcript level using "ChIPpeakAnno", "ChIPQC", and "ChIPseeker" packages of R. "DiffBind" package was used for differential binding.

H3K4me1, H3K4me3, H3K27ac and H3K27me3 ChIP data were extracted from GSE163548 (GEO repository).

## CUT&Tag analysis

For measurements of H3K27me3 after *HOTAIR* silencing by CUT&Tag, the CUT&Tag-IT™ Assay Kit (Active Motif) was used as recommended by the manufacturer. In brief, 200'000 knee SF (n = 2 OA) were seeded in 6 well plates and transfected with *HOTAIR* or control GapmeR as described above. After 48 h, cells were removed from the plate with trypsin (0.05%) and washed twice with wash buffer. Cells were bound to activated concanavalin A beads for 10 min at room temperature and then bound to the 1 µg of H3K27me3 antibodies (C15410195, Diagenode) overnight at 4 °C. Binding to the secondary antibody (guinea pick anti-rabbit) was done 60 min at room temperature. Immunoprecipitated samples were incubated with CUT&Tag-IT™ Assembled pA-Tn5 Transposomes for 60 min at room temperature, and then tagmented with Tagmentation buffer for 60 min at 37 °C. DNA was extracted and amplified with uniqe i7 and i5 index primers for each reaction. After clean up with SPRI beads, the libraries were sequenced on an Illumina NextSeq500. The reads were trimmed to 25 bp using Cutadapt (https://doi.org/10.14806/ej.17.1.200). The alignment to HG38 was performed using Bowtie2 with an average overall alignment rate 91.3%[84]. The reads with mapping quality lower than 2 were eliminated using samtools[83]. Peak calling was performed using SEACR by selecting the top 1% of regions by AUC[85]. Visualization of the results was performed using deepTools package[86].

## DNA methylation

Measurements of DNA methylation were extracted from previously published data[17]. In brief, DNA was isolated from hand (n = 1 OA/4 RA) and knee (n = 1 OA/4 RA) SF using the QIAamp DNA blood kit, Qiagen. DNA methylation was measured by Illumina HumanMethylation 450 array. Minfi was used to calculate β-values for each CpG site, COHCAP Bioconductor package (version 3.3) was used to calculate differentially methylated CpG sites. Differentially methylated CpG sites were calculated using analysis of variance F-statistic for averaged β-values (delta-β-value) per group of hand and knee samples. Average wig files for each anatomical location and the University of California Santa Cruz (UCSC) Genome Browser were used for data visualization.

## RNA sequencing

Total RNA was isolated with the miRNeasy Mini kit (Qiagen) including on-column DNaseI digestion from SF silenced for *HOTAIR* and control SF (n = 3 for each) 48 h after the transfection. RNA quantity and quality were evaluated using the Agilent RNA 6000 Nano kit with the Agilent 2100 Bioanalyzer instrument (Agilent Technologies, Inc.). The Illumina TruSeq Stranded total RNA protocol with the TruSeq Stranded total RNA Sample Preparation Kit was used to produce RNA-seq libraries. The quality and quantity of the generated libraries were determined by Agilent Technologies 2100 Bioanalyzer with DNA-specific chip and quantitative PCR (qPCR) using Illumina adapter-specific primers using the Roche LightCycler system (Roche Diagnostics), respectively. Diluted indexed long RNA-seq (10 nM) libraries were pooled in equal volumes, used for cluster generation (TruSeq SR Cluster Kit v3-cBot-HS reagents, according to the manufacturer's recommendations) and sequenced (TruSeq SBS Kit v3-HS reagents, Illumina HiSeq4000). Sequencing data reads were quality-checked with FastQC. Reads were trimmed with Trimmomatic and aligned to the reference genome and transcriptome (FASTA and GTF files, respectively, Ensembl GRCh37) with STAR[87]. Gene expression was quantified using the R/Bioconductor package Rsubread[88] version 1.22. Differentially expressed genes between conditions were identified using the R/Bioconductor packages DESeq2[89].

Pathway enrichment analyses of genes differentially expressed between SF invalidated for *HOTAIR* and controls were performed using Enrichr (all genes with FDR < 0.05, log fold change +/−1)[74]. The scatter dot plot was created with the Enrichr Appyter.

## ELISA

The human CXCL12 DuoSet ELISA kit (R&D Systems), the human IL12-p35 ELISA kit (Elabscience), the human IL6 ELISA DuoSet kit (R&D Systems), the human procollagen 1α ELISA Set (BD Biosciences), and the human FGF8 ELISA Kit (MyBioSource), respectively was used with cell culture supernatants.

## SF organ micromasses

3D micromasses were generated as previously described[90]. In brief, SF transfected with control or *HOTAIR* GapmeR were mixed with Matrigel (LDEV-free, Corning) (3 × 10^6 SF/ml Matrigel) and 30 µl droplets added to 12-well plates coated with poly 2-hydroxyethylmethacrylate (Sigma). Micromasses were left in culture for 3 weeks in Dulbecco's modified Eagle's medium (DMEM; Life Technologies) supplemented with 10% fetal calf serum (FCS), 1% penicillin/streptomycin, 1% minimum Essential medium non-Essential Amino Acids (Gibco), 1% ITS + premix (BD) und 17.6 µg/ml vitamin C. Medium was changed every week and supernatants for IL6 measurements were collected after the first week in culture. After 3 weeks, micromasses were fixed with 2% paraformaldehyde. After 24 h, paraformaldehyde was replaced by 70% ethanol and micromasses were embedded in paraffin and sectioned for IHC. Spheroids were stained with anti-human collagen I antibodies (ab138492, Abcam; dilution: 1:1000) on a BOND-MAX autostainer (Leica). Staining was imaged on a Zeiss Imager.Z1 (25x magnification) and quantified with ImageJ using brightness values.

## Western blotting

Cells were lysed in Laemmli buffer (62.5 mM TrisHCl, 2% SDS, 10% Glycerol, 0.1% Bromphenolblue, 5 mM β-mercaptoethanol). Whole cell lysates were separated on 10% SDS polyacrylamide gels and electroblotted onto nitrocellulose membranes (Whatman). Membranes were blocked for 1 h in 5% (w/v) non-fat milk in TBS-T (20 mM Tris base, 137 mM sodium chloride, 0.1% Tween-20, pH 7.6) or in 5% (w/v) BSA in TBS-T BSA for phosphorylated proteins. After blocking, the membranes were probed with antibodies against p-AKT (4060, Cell Signaling; dilution: 1:1000), AKT (4685, Cell Signaling; dilution: 1:1000) α-tubulin (ab7291, abcam; dilution 1:10000) overnight at 4 °C. As secondary antibodies, horseradish peroxidase-conjugated goat anti-rabbit (111-036-047, Jackson ImmunoResearch; dilution: 1:10'000) or goat anti-mouse antibodies (115-036-062, Jackson ImmunoResearch; dilution: 1:10'000) were used. Signals were detected using the ECL Western blotting detection reagents (GE Healthcare) and the Alpha Imager Software system (Alpha Innotech).

## Luciferase activity assay

To measure the effect of HOTAIR silencing on the canonical Wnt pathway, SF were transfected by electroporation (BTX) with the beta-catenin reporter M50 Super 8x TOPFlash (Addgene plasmid #12456) or M51 Super 8× FOPFlash, which contains mutated binding sites upstream of the luciferase reporter (Addgene plasmid #12457)[91]. Both plasmids were a gift from Randall Moon. For normalization, pRenilla Luciferase Control Reporter Vectors (Promega) were co-transfected. 24 h after transfection, cells were transfected with GapmeR for *HOTAIR* or control as mentioned above. Luciferase activity was measured with a dual luciferase reporter assay system (Promega), and the results were normalized to the activity of Renilla luciferase. Wnt signaling activation was also determined by LEF (lymphoid enchancer factor) Cignal Reporter assay (Qiagen). SF were transfected with the WNT reporter, negative control or positive control (GFP) plasmid, respectively using nucleofection (BTX, 0.2 cm cuvette, 180 V, 20mS). Each plasmid was co-transfected with GapmeR for *HOTAIR* or control. After 48 h, luciferase activity was measure with the dual luciferase reporter assay system (Promega).

### Real-time cell analysis (RTCA)

For RTCA of cell adhesion and proliferation of SF, the xCELLigence RTCA DP Instrument (ACEA Biosciences, Inc.) was used. 16-well E-plates were equilibrated with 100 µl of DMEM, 10% FCS for 30 min at RT. The impedance, expressed as arbitrary Cell Index (CI) units, of the wells with media alone (background impedance–Rb) was measured before adding the cells. SF were detached with accutase (Merck), resuspended in DMEM, and seeded at a cell density of 25,000 cells per well. Cell adhesion and spreading, measured as changes in impedance, was monitored every 5 min for a period of first 12 h and every 15 min after that for the next 12 h. The CI at each time point is defined as (Rn −Rb)/15, where Rn is the cell-electrode impedance of the well when it contains cells and Rb is the background impedance. Each condition was analysed in quadruplicates. Impedance changes were recorded every 15 min (0–24 h) and every 30 min (24–245 h). Adhesion was analysed over the first 16 h of the experiment, spreading was analysed between 17 and 127 h and proliferation during the exponential phase of the slopes (120-245).

### Scratch assay

To assess real time migration of SF, we performed a scratch assay[92] using Ibidi culture inserts (Ibidi, Germany) with 70 µl cell suspension (1.2 × 10^5 cells/ml). Cells were incubated at 37 °C and 5% CO$_2$ for 24 h to obtain a confluent cell layer. Next, cells were transfected with GapmeR *HOTAIR* or controls and incubated for additional 24 h. Culture inserts were removed and cell layers were washed with PBS. Culture medium (with vitamin C) was added and time lapse images were recorded every 30 min for 48 h using a widefield Zeiss AxioObserver equipped with a stage incubator to maintain temperature and CO$_2$ conditions. Each assay was performed in triplicate and repeated three times. Open area was calculated over 24 h to 48 h according to experiments[92].

### Apoptosis assay

Activity of the key effector caspases 3 and 7 was measured using the Caspase-Glo 3/7 assay (Promega, Madison, WI). SF were seeded at a density of 4000 cells/well in 96-well white-walled plates. The next day, SF were transfected with GapmeR for *HOTAIR* or GapmeR control or left untransfected. Cleaved caspase 3/7 activity was assessed 48 h after transfection. FAS-ligand (2 µg/ml) was added 18 h before the assay to stimulate apoptosis. Luminescence signals were measured using a Synergy HT microplate reader (Bio Tek). Each assay was performed in triplicate and repeated two times.

### Osteoclastogenesis assay

Human osteoclasts precursors were isolated from blood donations of healthy volunteers ($n = 4$, Red Cross, Schlieren, Switzerland). In brief, CD14+ cells were isolated by positive selection using magnetic separation (Milteny Biotec) after a Ficoll gradient (GE Healthcare) separation. Isolated monocytes (5 × 10^4) were cultured in chamber slides in αMEM supplemented with 10% FBS (GE Healthcare), 2 mM l-glutamine and antibiotics in the presence of 25 ng/mL macrophage colony-stimulating factor (M-CSF) (PeproTech) for 3 days. SF were transfected with *HOTAIR* or control GapmeR and were added to differentiating monocytes (5 × 10^5 SF/well). Alternatively, supernatants from control or *HOTAIR* silenced SF were added. Cells were cultured in the presence of 25 ng/mL M-CSF, 50 ng/mL RANKL (PeproTech) and vitamin C. The media was replaced every 2 days. After 6 days of culture, the cells were fixed by 4% paraformaldehyde and were stained by TRAP. Control experiments included untransfected SF, SF alone (without precursors of osteoclasts) and precursors without SF or supernatants. Each assay was performed in duplicate and repeated three times.

To assess bone resorption, osteoclasts precursors (10^5/well) were co-cultured with SF (1.5 × 10^4/well) or supernatants from SF in 96 wells on bovine bone slices (Jelling, Denmark) for two weeks in the presence of 25 ng/mL M-CSF, 50 ng/mL RANKL and vitamin C. The osteoclastogenic medium was replaced every 2–3 days. SF were re-transfected every 5 days with *HOTAIR* or control GapmeRs. Analyses were done after 14 days of differentiation. Cells on bone slices were subsequently incubated with a 0.1 m NaOH solution, ultrasonicated for 2 min, rinsed with water to remove cells from the slices, and placed in a 1% aqueous toluidine blue solution containing 1% sodium borate for 5 min. Photomicrographs of resorption pits were taken using a light microscope. Resorption area was measured using Image J.

### Chemotaxis assay

PBMCs from one healthy donor were isolated using CPT™ tubes (BD Biosciences) and 10^6 PBMCs were seeded in the upper well of a 96-well transwell migration chamber with 5 µm pore size (Corning). The lower chamber was filled with supernatants collected from control or *HOTAIR* silenced SF 48 h after transfection (triplicates). Unconditioned medium or PBS were used as controls. Cells were collected from the lower chamber after 18 h using ice-cold 20 mM EDTA/0.5% FCS in PBS as detachment solution. Collected cells were counted in a cell counter (Casy, OLS) and stained with anti-human CD14-PE (Clone REA599, Miltenyi; 1 µl; 1:200), anti-human CD19-PE (Clone HIB19, Invitrogen; 1 µl; 1:200) or anti-human CD3-FITC (clone UCHT1, Invitrogen; 1 µl; 1:200) antibodies for 1 h at 4 °C. FITC- and PE-labelled IgG were used as negative control antibodies. Percentage of positive cells was assessed on a FACSCalibur™ flow cytometry platform (BD Biosciences). Gates were set so that <2% positive cells were detected in the IgG stained cells (Supplementary Fig. S6).

### RNA sequencing from the early arthritis cohort

RNA-sequencing data from the Pathobiology of Early Arthritis Cohort, which is available on PEAK website (https://peac.hpc.qmul.ac.uk/) was studied. RNA-seq data have been deposited by the owners of the data (Christopher John and Myles Lewis, Queen Mary University of London) in ArrayExpress under Accession code E-MTAB-6141[93].

### Isolation of mouse tissue from lung, spine and gut

Wild-type C57BL/6 mice, 6 weeks old, were purchased from Jackson and dissected. The spine and gut were isolated and separated into different parts (spine: cervical, thoracic and lumbal) and gut (stomach, small intestine, caecum, colon and rectum). Total RNA was isolated using the miRNeasy Mini kit (Qiagen) including on-column DNaseI digestion. Total RNA was reversed transcribed and qPCR was performed using SYBR green (Life Technologies). Constitutively expressed *beta2-microglobulin* was measured for internal standard sample normalization. In cases of undetectable expression of *Hotair* in mouse tissue*s* the ct was set arbitrarily to 45 in order to calculate the dct.

### Statistical analysis

Data were analysed with GraphPad Prism version 6.0 or higher, IBM SPSS Statistics software and R version 4.3.1. Normality was tested using Shapiro–Wilk test. Two groups were compared with two-tailed unpaired or paired t test, Mann Whitney test, Wilcoxon matched-paired, one sample t test or one sample Wilcoxon, as appropriate. Multiple group comparisons were performed by adjustments for multiple comparisons using Bonferroni correction or two-way ANOVA. Correlations were tested using Pearson's correlation coefficient. All tests were performed two-sided, if not otherwise indicated. $P$ values of <0.05 were considered statistically significant. All experiments stem from at least two technical repeats. All replicates are biological replicates.

### Study approval

The collection and experimental usage of the human samples was approved by the ethical commission of the Kanton Zurich (swissethics number: 2019-00674, PB-2016-02014 and 2019-00115). Informed

consent was obtained from all patients. All experiments have been performed in accordance with the institutional guidelines.

Mouse experiments were approved by the Institutional Committee of Protocol Evaluation in conjunction with the Veterinary Service Management of the Hellenic Republic Prefecture of Attika according to all current European and national legislation and were performed in accordance with relevant guidelines and regulations, under the relevant animal protocol licenses with number 2199-11/4/2017.

### Reporting summary

Further information on research design is available in the Nature Portfolio Reporting Summary linked to this article.

## Data availability

The RNA seq, ChIP seq of H3K27me3 and the scRNA seq data with control GapmeR and *HOTAIR* GapmeR transfected SF have been deposited in the GEO repository under accession code GSE185440. The scRNAseq data from synovial tissues have been deposited in the Array Express repository under accession code E-MTAB-11791. The CAGE and the H3K4me1, H3K4me3, H3K27ac and H3K27me3 ChIP data were obtained from the GEO repository under the accession number GSE163548. RNA-sequencing data from the Pathobiology of Early Arthritis Cohort is available on the PEAK website https://peac.hpc.qmul.ac.uk/ and in the Array Express repository under the accession code E-MTAB-6141. Source data are provided with this paper.

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

## Acknowledgements

We thank Maria Comazzi and Peter Künzler (Center of Experimental Rheumatology, University Hospital Zurich, Switzerland) for excellent technical assistance. We thank Miriam Marks for providing the samples from joint replacement. We thank the Functional Genomics Center Zurich for excellence support in sequencing. ME was supported by Société Française de Rhumatologie, Fondation pour la Recherche Médicale, Assistance publique - Hôpitaux de Paris (APHP) and EULAR. CO was supported by the Hartmann-Müller Foundation (2238), the Foundation for Research in Science and the Humanities at the University of Zurich (STWF-19-010), the EMDO Foundation (1011), the Iten-Kohaut Foundation, the Novartis Foundation for medical-biological Research (20A012) and by the IMI project BTCure (GA no. 115142-2). MA, GK and MS acknowledge support by the InfrafrontierGR infrastructure (MIS 5002135), co-funded by Greece and the European Union [European Regional Development Fund] under NSRF 2014–2020, which provided mouse hosting and phenotyping facilities.

## Author contributions

C.O., M.E. and R.M. designed, analyzed and interpreted experiments. M.E., C.O., R.M. and M.H. wrote the manuscript. C.O., M.E., M.T., L.K., M.H., M.M, L.M., C.P., K.B., K.K., P.S., M.F.B. and S.G.E. performed experiments. R.M., A.K., C.G. and M.H. did the computational analysis. M.E., A.L., G.Kania., M.S., G.Kollias, M.A. and C.O. performed the mouse experiments. R.M., K.B., T.R. and O.D. organized ethic approval, recruited patients and collected samples. All authors critically reviewed the manuscript.

## Competing interests

M.E., R.M., L.K., A.K., M.T., T.R., C.G., M.H., M.M., L.M., C.P., K.B., A.L., G.K., K.K., P.S., M.F.B., S.G.E., M.S., G.K., M.A., and C.O. declare no competing interests. O.D. has/had relationships with the following companies in the area of potential treatments for systemic sclerosis and its complications in the last three calendar years: Speaker fee: Bayer, Boehringer Ingelheim, Janssen, Medscape Consultancy fee: 4P-Pharma, Abbvie, Acceleron, Alcimed, Altavant Siences, Amgen, AnaMar, Arxx, AstraZeneca, Baecon, Blade, Bayer, Boehringer Ingelheim, Corbus, CSL Behring, Galapagos, Glenmark, Horizon, Inventiva, Kymera, Lupin, Miltenyi Biotec, Mitsubishi Tanabe, MSD, Novartis, Prometheus, Redxpharna, Roivant, Sanofi and Topadur Research Grants: Kymera, Mitsubishi Tanabe, Boehringer Ingelheim Oliver Distler has/had relationships with the following companies in the area of potential treatments for dermatomyositis and its complications in the last three calendar years: Consultancy fee for rheumatology topic: Pfizer (2021) Oliver Distler has/had relationships with the following companies in the area of potential treatments for arthritides in the last three Consultancy fee: Abbvie.
