## [Peer Review File · Nature Communications]

The long non-coding RNA HOTAIR contributes to joint-specific gene expression in rheumatoid arthritisREVIEWER COMMENTS

Reviewer #1 (expert in rheumatoid arthritis):

This is an interesting study from a well-established group evaluating the role of HOTAIR as a determinant of joint-location specific biology. The study builds on several previous publications showing that epigenetic marks and the transcriptome vary between joints, especially for fibroblasts.

1. Introduction: The use of the term pathotype is increasingly viewed as problematic, in part because that is a microbiology term that is not related to histology. More precise terms, such as histologic pattern or the recent AMP classification as "CTAPs" could be used. The review of joint specific findings in the introduction focuses solely on the authors' work and should be broadened to highlight some of the studies published prior to their excellent work, including differential marks and expression of various HOX genes.

2. Joint differences in histology were quite interesting in Figure 1. Clinical correlation would be important, such as a correlation between the degree of histologic synovitis in individual patients. The differences in the cell populations between the joints was difficult to interpret and subtle in Figure 1D because it is underpowered with only 4 knee tissues. Additional knee samples should be evaluated to determine if the differences are significant and reproducible. Similarly, the fibroblast studies in Figure 1 also should include more samples. There is also a concern that only 2 of the 4 knee and 5 of the 8 hand samples were from seropositive RA. Because seronegative RA is likely a distinct disease, a more homogeneous study population with more samples would be important. Figure 1G is very difficult to understand and a supplementary table might be more appropriate.

3. Figure 2 shows statistical significance for synovial HOTAIR, but in Figure 2D it is driven mainly by a couple of samples. In addition, the qPCR studies are on homogenized tissue. The authors note that expression is mainly in fibroblasts, and so the differences could be due to variable numbers of fibroblasts in the tissues and diseases.

4. The authors present some data indicating that HOTAIR regulates some histone modifications, but the conclusions are based on only in two knee cell lines. Additional lines would need to be studied to confirm because there is substantial line to line differences in synovial fibroblast epigenetic marks.

5. For silencing experiments, how long is HOTAIR depleted? The micromass experiments last at least 3 weeks, and presumably the authors have information confirming that knockdown is persistent.

6. In Figure 4, several subsections, like C, D, and F and not convincing. That is probably because the analysis was paired but that is not depicted in the bar graphs. These could be replotted to make it clearer. Some of the experiments have very few samples, such as Figure 4K.

7. The studies in transcriptionally defined synovial fibroblasts focuses on 4 populations, although the ones selected do not include key functional subsets. For example, phenotypes distinguished by CD90 expression and the microenvironment-related milieu, particularly NOTCH signaling, are probably more relevant to disease pathogenesis and HOTAIR biology. Figure 5C is does not convincingly show that the fibroblast populations are determined or regulated by HOTAIR because the differences are very small.

8. Many of functional studies, including migration and fibroblast-supported osteoclastogenesis, are largely confirmatory of studies in other fibroblast populations. Are there some functions that are unique to synovial fibroblasts showing differences related to joint location?

9. The correlation in Figure 7I ($r^2=0.27$) is weak and it is difficult to draw conclusions.

10. Several critical questions remain unanswered that should be experimentally addressed, including:

a. Is the phenotype of a hand vs. knee fibroblast fixed or can it be modulated or converted by the environment similar to the NOTCH3 effects in synovium. If it is fixed, where does the imprinting occur?

b. What is the mechanism of differential expression of HOTAIR in hand vs knee fibroblasts?

c. The authors focus on hand vs knee joints and ascribe joint specific biology to HOTAIR. However, there is also a literature (including from the authors) on differences between other joints. This is particularly true for hip vs knee joints where substantial differences in HOX gene expression occur even though the differences in HOTAIR expression are minimal. This suggests that there are other factors that determine the local transcriptome and fibroblast phenotype. How do the authors proposed to address that?

Reviewer #2 (expert in non-coding RNA):

The manuscript by Elhai M et al., describes the role of HOTAIR in inflammation in chronic arthritis. Specifically, the Investigators present data on joint-specific molecular and tissue changes in the synovium and in synovial fibroblasts. They show that TNF-dependent down-regulation of HOTAIR leads to reduced osteoclastogenesis and increased B cell recruitment.

One of the fundamental limitations of this manuscript is that the Authors do not provide any evidence of the degree of silencing achieved with their Gapmer designed to silence HOTAIR. Besides, no Gapmer will be able to completely silence any gene expression, so a more suitable approach would be a CRISPR-based silencing, which is an essential missing approach to all the key experiments in this study.

Another concern is about the lack of rigor in data interpretation. The Authors do not report p values in experiments that are described as significant changes (e.g. Fig. 4E, 4G, 4I, 7D, 7E, 7G). Moreover, COL3A1, commented on the main text on page 13, is not reported in Fig. 4B. Another problem is expressing variations of gene mRNA levels, without looking whether such variations also occur at the protein level (e.g. but not limited to FGF8 expression).

Also, in Fig. 3E why was the 48h timepoint chosen? And why did the Authors look at variations of phosphor AKT after 72h from HOTAIR silencing (Fig. 4F). There is no explanation of the rationale behind the choice of these timepoints over others and why these differences between one experiment and another.

Moreover, on page 14 the authors refer to IL12, but in the figure it's reported IL12A. So, why didn't they mention IL12A also in the main manuscript? On the same page, it is totally unclear why CTNNB1, FGFR2 and LGR5 were chosen.

Fig. 6F and 6G, right sides: what is the difference between black and white dots? A proper legend is missing.

Fig. 7G: are the reported average values a combination of RA and OA samples? If so, why weren't they presented separately for the two different conditions?

Page 18: the conclusion that "could be an effect of decreased secretion of CXCL12 by HOTAIR silenced SF" is absolutely random and not supported by any experimental evidence.

Supplemental Table 2 appears cut (at least in the title) and cannot be accepted in the present format.

Response to reviewers:

We thank the reviewers for thoroughly reviewing our manuscript and their useful comments, which helped us to improve the quality of the manuscript significantly.

For the revision, we needed the help of various people who have been added to the list of co-authors in our revised manuscript (Leandra Keusch, Alexandra Khmelevskaya, Melpomeni Toitou, Thomas Rauer and Celina Geiss).

Please, see our point-to-point replies to the concerns raised by the reviewers below.

REVIEWER COMMENTS

Reviewer #1 (expert in rheumatoid arthritis):

Reviewer comment: This is an interesting study from a well-established group evaluating the role of HOTAIR as a determinant of joint-location specific biology. The study builds on several previous publications showing that epigenetic marks and the transcriptome vary between joints, especially for fibroblasts.

1. Introduction: The use of the term pathotype is increasingly viewed as problematic, in part because that is a microbiology term that is not related to histology. More precise terms, such as histologic pattern or the recent AMP classification as “CTAPs” could be used.

Response: As suggested by the reviewer, we changed pathotype in “histological pattern” throughout the revised manuscript.

Reviewer comment: The review of joint specific findings in the introduction focuses solely on the authors’ work and should be broadened to highlight some of the studies published prior to their excellent work, including differential marks and expression of various HOX genes.

Response: We fully agree with the reviewer and are sorry that we missed important other work in this field. We have broadened the introduction in the revised manuscript with the following sentences

Page 6 line 1:

‘Joint specific patterns of DNA methylation in HOX genes were found in SF (18) as well as in cartilage (19) which suggested an embryonically imprinted joint specific stromal signature.’

Page 6 line 7:

‘During embryogenesis, the expression of specific HOX genes delineates distinct positional identities that lead to specific cell differentiation and tissue morphogenesis (20).

Interestingly, it has been shown that also adult skin fibroblasts maintained key features of HOX gene expression patterns established during embryogenesis (23) and also in skin fibroblasts site-specific HOX expression is epigenetically maintained (20, 24).’

We added the following references:

18. R. Ai, D. Hammaker, D. L. Boyle, R. Morgan, A. M. Walsh, S. Fan, G. S. Firestein, W. Wang, Joint-specific DNA methylation and transcriptome signatures in rheumatoid arthritis identify distinct pathogenic processes. *Nat Commun* 7, 11849 (2016).

19. W. den Hollander, Y. F. Ramos, S. D. Bos, N. Bomer, R. van der Breggen, N. Lakenberg, W. J. de Dijcker, B. J. Duijnsveld, P. E. Slagboom, R. G. Nelissen, I. Meulenbelt, Knee and hip articular cartilage have distinct epigenomic landscapes: implications for future cartilage regeneration approaches. *Ann Rheum Dis* 73, 2208-2212 (2014).

23. H. Y. Chang, J. T. Chi, S. Dudoit, C. Bondre, M. van de Rijn, D. Botstein, P. O. Brown, Diversity, topographic differentiation, and positional memory in human fibroblasts. *Proc Natl Acad Sci U S A* 99, 12877-12882 (2002).

24. J. L. Rinn, J. K. Wang, N. Allen, S. A. Brugmann, A. J. Mikels, H. Liu, T. W. Ridky, H. S. Stadler, R. Nusse, J. A. Helms, H. Y. Chang, A dermal HOX transcriptional program regulates site-specific epidermal fate. *Genes Dev* 22, 303-307 (2008).

Reviewer comment: 2. Joint differences in histology were quite interesting in Figure 1. Clinical correlation would be important, such as a correlation between the degree of histologic synovitis in individual patients.

Response: As suggested by the reviewer, correlation between the degree of histologic synovitis and clinical characteristics were performed in the revised manuscript. We did not observe any correlation between the degree of histological synovitis and CRP or DAS28ESR in individual patients.

This has been added to our revised manuscript in Supplemental Figure S1 and in page 7 lines 8-9: 'There was no association of the Krenn synovitis score with clinical characteristics of the disease in individual patients such as CRP or DAS28ESR ($p=0.71$ and $p=0.48$, respectively) (Supplementary Figure S1 A-B).'

Reviewer comment: The differences in the cell populations between the joints was difficult to interpret and subtle in Figure 1D because it is underpowered with only 4 knee tissues. Additional knee samples should be evaluated to determine if the differences are significant and reproducible.

Response: We were able to increase the sample number of knee synovial tissues to $n = 6$. To improve the interpretation of the proportion plots in Figure 1D, we added boxplots of the individual cell populations in hand and knee synovial tissues (supplementary Figure 1C) and modified the text in the manuscript. Page 7, line 18: 'Consequently, analysis of synovial cell proportions by single cell RNA sequencing (scRNAseq) showed expansion of T- cell compartments in wrist compared to knee RA synovium and higher proportions of myeloid cells in knee versus wrist synovium (Figure 1D, Supplementary Figure S1C and Table 1).'

Reviewer comment: Similarly, the fibroblast studies in Figure 1 also should include more samples. There is also a concern that only 2 of the 4 knee and 5 of the 8 hand samples were from seropositive RA. Because seronegative RA is likely a distinct disease, a more homogeneous study population with more samples would be important.

Response: Since the sample number of knee synovial tissues for scRNAseq was increased, we now have $n = 6$ for the fibroblast analysis. Four out of the 6 patients were seropositive, for one we have missing information and one was seronegative. We agree that seronegative RA is likely a distinct disease. Therefore, to address this issue, we performed the same analysis including only samples from seropositive RA. 83% and 87% of the genes upregulated in hand and knee synovial fibroblasts, respectively from all RA patients were also increased when only synovial fibroblasts from seropositive RA patients were used for the analysis. Therefore, we think that in this dataset the influence of differences between seropositive and seronegative RA on joint specific gene expression is minor.

Reviewer comment: Figure 1G is very difficult to understand and a supplementary table might be more appropriate.

Response: We completely agree with the reviewer that the presentation of the data in Figure 1G was not very comprehensive. We now changed the Venn diagram to an

Upset plot showing the number of shared genes between the analysed conditions. We hope the reviewer agrees that with this representation it is clearer to see that HOTAIR regulates most of the differentially expressed genes between knee and hand synovial fibroblasts.

Reviewer comment: 3. Figure 2 shows statistical significance for synovial HOTAIR, but in Figure 2D it is driven mainly by a couple of samples.

Response: We thank the reviewer for this relevant comment. During the revision process, we could increase the sample size to 11 OA knees and 9 RA knees. Although there is some heterogeneity, the results confirm a higher expression of HOTAIR in OA as compared to RA. The results are presented in revised Figure 3D and the characteristics of the patients in the revised manuscript in Table 2.

Reviewer comment: In addition, the qPCR studies are on homogenized tissue. The authors note that expression is mainly in fibroblasts, and so the differences could be due to variable numbers of fibroblasts in the tissues and diseases.

Response: We thank the Reviewer for noting this relevant point. To ensure that the difference observed was indeed linked to a difference in HOTAIR expression according to diseases and not to a difference in the number of fibroblasts, we normalized HOTAIR expression of 7 RA and 7 OA samples on a gene specific to fibroblasts, namely COL3A1 and were able to show that the results were not changed. In our opinion, this confirmed the difference in HOTAIR expression between OA and RA synovial tissues. This additional experiment and the results have been added to our revised manuscript in Figure 3F and page 12 line 10: 'To ensure that the difference observed was indeed linked to a difference in HOTAIR expression according to diseases and not to a difference in the number of fibroblasts in the tissue, we normalized the expression of HOTAIR in the tissue to a fibroblast specific gene, namely COL3A1. This approach did not change the result (Figure 3F), confirming the difference in HOTAIR expression between OA and RA in vivo.'

Reviewer comment: 4. The authors present some data indicating that HOTAIR regulates some histone modifications, but the conclusions are based on only in two knee cell lines. Additional lines would need to be studied to confirm because there is substantial line to line differences in synovial fibroblast epigenetic marks.

Response: Modulation of H3K27me3 by HOTAIR is well described by several previous studies, in particular in skin fibroblasts (see also the cited articles Tsai et al. Long noncoding RNA as modular scaffold of histone modification complexes. Science, 2010 and Rinn et al. Functional demarcation of active and silent chromatin domains in human HOX loci by noncoding RNAs. Cell, 2007). Therefore, we only performed ChIP in two cell lines to confirm this regulation in synovial fibroblasts. However, we agree that it would improve the quality of our data to further confirm modulation of H3K27me3 by HOTAIR in synovial fibroblasts. Because of the high numbers of cells needed, ChIP experiments are challenging in primary synovial fibroblasts. Therefore, and to confirm the result with another method, we performed CUT&Tag analysis in 2 additional OA synovial fibroblast samples silenced for HOTAIR and saw the same result. Heatmaps of enriched H3K27me3 in control silenced and HOTAIR silenced synovial fibroblasts are shown in Supplementary Figure S2B. The following text was added to the manuscript on page 14 line 9:

We could confirm these data by using a CUT&Tag approach, which also showed decreased levels of H3K27me3 enrichment in SF after HOTAIR silencing (Supplementary Figure S2B).

Reviewer comment: 5. For silencing experiments, how long is HOTAIR depleted? The micromass experiments last at least 3 weeks, and presumably the authors have information confirming that knockdown is persistent.

Response: We apologize that we missed to include this information in the original version of the manuscript. We now added the time course experiment showing that HOTAIR is still depleted after one week but levels are back to normal after 10 days. This has been added to our revised manuscript in Supplementary Figure S2A. We also realized that there was missing information in the description of the experiments performed with the micromasses. IL6 was measured in the micromass supernatant after one week in culture without medium change. This is now specified in the Methods section (page 41, line 7: 'Medium was changed every week and supernatants for IL6 measurements were collected after the first week in culture.'). To allow the micromass to grow and the extracellular matrix to accumulate, we waited three weeks to stain the micromasses for collagen. Although HOTAIR is no longer depleted by this time, we believe that the first week in culture in the absence of HOTAIR induced enough differences in collagen production to see the effect still after 3 weeks. This is also supported by the fact that elimination of HOTAIR resulted in significant changes in H3K27me3 marks.

Reviewer comment: 6. In Figure 4, several subsections, like C, D, and F and not convincing. That is probably because the analysis was paired but that is not depicted in the bar graphs. These could be replotted to make it clearer.

Response: As suggested by the reviewer, paired experiments have been replotted in Figure 5 in the revised manuscript. We agree that not in all measurements the differences are equally strong and homogenous, as it is often the case when working with primary human cells. For instance, IL6 production is highly heterogeneous between patients, which has previously been acknowledged in synovial fibroblasts and explained by differences in the genetic background (Noss et al. Genetic polymorphism directs IL-6 expression in fibroblasts but not selected other cell types, PNAS, 2015). We now also added additional time points to the measurements to illustrate the evolution of the response over time.

Reviewer comment: Some of the experiments have very few samples, such as Figure 4K.

Response: To strengthen the reliability of the data we added more samples to the data presented in Figures 1D-G, 3D, 5K, 5L. Furthermore, we repeated the analysis of changes of H3K37me3 after HOTAIR silencing with a CUT&tag approach (Supplementary Figure S2B) and the analysis of WNT signaling after HOTAIR silencing with a commercially available LEF reporter assay (Figure 5I).

Reviewer comment: 7. The studies in transcriptionally defined synovial fibroblasts focuses on 4 populations, although the ones selected do not include key functional subsets. For example, phenotypes distinguished by CD90 expression and the microenvironment-related milieu, particularly NOTCH signaling, are probably more relevant to disease pathogenesis and HOTAIR biology. Figure 5C does not convincingly show that the fibroblast populations are determined or regulated by HOTAIR because the differences are very small.

Response: We thank the reviewer for this comment and agree that it is important to focus on previously defined functional SF subsets. Therefore, we re-run the analysis including key functional subsets of SF and still observed significant changes in the proportions of SF between HOTAIR-silenced and control SF. Although statistically significant overall, these changes were indeed small and should not be

overinterpreted. We have reformulated the conclusions in this part of the study accordingly. However, we still find it interesting that HOTAIR may differentially regulate some marker genes such as COL1A1 and CXCL12 in SF subtypes and suppress these transcripts in one SF subtype but promote their expression in another. The new analysis is shown in Figure 6, Supplementary Figure S4 and described starting from page 18 line 21:’ We recapitulated nine key functional subpopulations of SF that have been described previously by scRNAseq analysis in synovial tissues (Figure 6A, Supplementary Figure S4A). CD90-/PRG4+ SF are considered as lining SF, whereas CD90+ subpopulations are located in the sublining. The RSP03+ SF population was described as intermediate lining/sublining phenotyp⁴⁴. In line with published data¹⁴, several of the in vivo SF subtypes were partially lost during culture (Figure 6B). The main SF subpopulations in culture were POSTN+ SF, CD34+ SF, NOTCH3+ cells and a mixed-marker cell population consisting of proliferating cells (prolif. SF) (Figures 6A/B). Silencing of HOTAIR was associated with a small but significant change in the overall proportions of SF subpopulations (Figure 6C). Individually, none of the changes in the SF subgroups reached statistical significance (Supplementary Figure S4B). However, there was a noticeable decrease in the proportion of proliferative SF and CD74hi SF in SF silenced for HOTAIR. POSTN+ SF and CXCL12+ SF were rather increased in the HOTAIR-silenced condition, even though CXCL12 transcription itself was decreased in this subtype, which is the main producer of CXCL12 (Figure 6C and Supplementary Figure S4A/B). In contrast to IL-6, which was upregulated in all SF subtypes, but similar to CXCL12, HOTAIR regulated COL1A1 and COL1A2 expression different across SF subtypes (Figure 6D). COL1A1 and COL1A2 were mainly increased in PRG4+ CLIC5+ SF after silencing, with no change or a decrease in the main collagen producing subtype, POSTN+ SF (Figure 6D and Supplementary Figure S4A). This might reflect a transcriptional switch from POSTN+ SF to collagen producing PRG4+ SF and might explain the inconsistent results seen in the bulk analysis of COL transcripts (Figure 5B). From this analysis, it can be hypothesized that HOTAIR could play a role in the formation of SF subtypes, for example, by directing differential collagen or CXCL12 production between the SF subtypes. However, there were also regulatory mechanisms of HOTAIR that were evident in all SF subtypes.’

Reviewer comment: 8. Many of functional studies, including migration and fibroblast-supported osteoclastogenesis, are largely confirmatory of studies in other fibroblast populations. Are there some functions that are unique to synovial fibroblasts showing differences related to joint location?

Response: Although we believe that differential expression of HOTAIR and other HOX genes between joints may shape some basic SF functions that could be important for the respective joint sites, such as collagen production and synovial fibroblast response to inflammatory and possibly other stimuli, we do not believe that there are synovial fibroblast functions that are exclusive to one joint site but not the other.

Reviewer comment: 9. The correlation in Figure 7I ($r^2=0.27$) is weak and it is difficult to draw conclusions.

Response: We agree with the reviewer, that the correlation is weak, but we think it has to be taken into consideration that the analysis was not a result of HARKing, but based on our results from the previous experiments. Data in Figures 8F, 8G and 8H show consistent results and also confirmed our hypothesis, which was based on the chemotaxis assays. We therefore believe that conclusions should be drawn in the light of Figure 8 as a whole. We now highlight this on page 22 line 28:’Although the correlation was weak, this result was supported by the previous findings in Figure 8F-H.’

Reviewer comment: 10. Several critical questions remain unanswered that should be experimentally addressed, including:

a. Is the phenotype of a hand vs. knee fibroblast fixed or can it be modulated or converted by the environment similar to the NOTCH3 effects in synovium. If it is fixed, where does the imprinting occur?

b. What is the mechanism of differential expression of HOTAIR in hand vs knee fibroblasts?

Response: We thank the reviewer for bringing up this important questions. We have further studied the mechanisms responsible for the differential expression of HOTAIR in hands and knees showing different epigenetic marks active in knee and hands. These data have been added to our revised manuscript in Figure 2 and the following text has been added (page 10 starting from line 6):

'We now further assessed the epigenetic mechanisms responsible for differential expression of HOTAIR between hand and knee SF. Analysis of the chromatin landscape of cultured RA and OA SF showed that hand SF carried repressive H3K27me3 marks broadly across the promoter and transcription start site of HOTAIR and did not show any active promoter or enhancers as assessed by Cap Analysis of Gene Expression (CAGE-seq) (Figure 2A-B). Knee SF used a different enhancer for HOTAIR expression (Figure 2A), which was also shown to be active by CAGE-seq (Figure 2B). Furthermore, knee SF had increased levels of activating H3K27ac marks in particular over the promoter region compared to hand SF (Figure 2A). Accordingly, HOTAIR expression in cultured knee SF was decreased after silencing of the histone acetyltransferases p300 and CBP (Figure 2C/D). In addition, we observed higher DNA methylation in the HOTAIR gene body in cultured SF from knees as compared to hands (Figure 2E). DNA de-methylation by 5-azacytidine further induced HOTAIR expression in knee SF without changes in hand SF (Figures 2F), suggesting that DNA methylation is an additional level of regulating HOTAIR gene expression in knee SF. Together these data confirm the existence of a strong, mitotically stable epigenetic imprinting of joint-specific HOTAIR expression in SF and suggest different levels of epigenetic control with histone modifications regulating joint-specific expression of HOTAIR and DNA methylation regulating HOTAIR expression in knee SF.'

Reviewer comment: c. The authors focus on hand vs knee joints and ascribe joint specific biology to HOTAIR. However, there is also a literature (including from the authors) on differences between other joints. This is particularly true for hip vs knee joints where substantial differences in HOX gene expression occur even though the differences in HOTAIR expression are minimal. This suggests that there are other factors that determine the local transcriptome and fibroblast phenotype. How do the authors proposed to address that?

Response: We absolutely agree with the reviewer that other factors such as other HOX genes, local mechanical or anatomical differences shape the joint stroma. As shown in Figure 1E and 1G, many other HOX genes are differentially expressed between hand and knees and are also involved in the local transcriptome and fibroblast phenotype. We are currently investigating the function of several of them in gene expression and fibroblast function (e.g. HOXD10-13, HOTTIP).

We now highlight this limitation in the discussion section of our revised manuscript (page 27 line 21): *'Moreover, we acknowledge that HOTAIR is not the only regulator of the differential transcriptome between hands and knees. Other HOX genes are differentially expressed between joints as highlighted in our Figure 1E and G. Their effect on the transcriptome and fibroblast function will need to be investigated in further studies. Furthermore, different mechanical and anatomical influences between joints might have a great impact on joint specific gene expression.'*

Reviewer #2 (expert in non-coding RNA):

Reviewer comment: The manuscript by Elhai M et al., describes the role of HOTAIR in inflammation in chronic arthritis. Specifically, the Investigators present data on joint-specific molecular and tissue changes in the synovium and in synovial fibroblasts. They show that TNF-dependent down-regulation of HOTAIR leads to reduced osteoclastogenesis and increased B cell recruitment.

One of the fundamental limitations of this manuscript is that the Authors do not provide any evidence of the degree of silencing achieved with their Gapmer designed to silence HOTAIR.

Response: We sincerely apologize that it slipped our attention that we did not include this figure in our initial manuscript. We have now included a time-course of HOTAIR silencing confirming more than 95% of silencing after 2 days (Figure S4A).

Reviewer comment: Besides, no Gapmer will be able to completely silence any gene expression, so a more suitable approach would be a CRISPR-based silencing, which is an essential missing approach to all the key experiments in this study.

Response: We fully agree with the reviewer that CRISPR-based silencing is preferable to GapmeR transfection when 100% silencing and long-term effects are to be studied. However, we think that our approach better mimics the effect of HOTAIR downregulation in a pro-inflammatory environment, which we wanted to study (in our experiments, TNF stimulation resulted in an average 60% reduction in HOTAIR expression). Therefore, we strongly believe that the results of our approach are highly relevant and show how pro-inflammatory cytokines can modulate arthritis-relevant pathways and functions via regulation of a joint-specifically expressed long non-coding RNA.

We included this point in the revised manuscript as follows (page 27 line 17): 'Another limitation is, that by using GapmeR technology, we did not obtain complete and long-lasting silencing of HOTAIR as could be achieved with CRISPR-based technology, for example. However, as our aim was to mimic the situation in vivo in a pro-inflammatory environment, i.e. a decrease in HOTAIR expression without complete silencing, the use of GapHOTAIR appeared to be the most suitable approach.'

Reviewer comment: Another concern is about the lack of rigor in data interpretation. The Authors do not report p values in experiments that are described as significant changes (e.g. Fig. 4E, 4G, 4I, 7D, 7E, 7G).

Response: We sincerely apologize for this error. All p-values ≤ 0.08 have been added to the figures with a note in the legend that the p-values > 0.08 do not appear in the figure.

Reviewer comment: Moreover, COL3A1, commented on the main text on page 13, is not reported in Fig. 4B.

Response: We thank the reviewer for detecting this mistake. The measurements of COL3A1 have been added to Figure 5B of our revised manuscript.

Reviewer comment: Another problem is expressing variations of gene mRNA levels, without looking whether such variations also occur at the protein level (e.g. but not limited to FGF8 expression).

Response : We agree with the reviewer that changes of mRNA level should be confirmed on protein level. However, we think that by confirming changes in several proteins (IL6, IL12, CXCL12, Collagen, and now also FGF8), predicted pathways (WNT,

AKT) and predicted functional changes (migration, apoptosis, osteoclastogenesis, chemotaxis), we have ensured that our data from RNA sequencing have functional relevance. We hope that the reviewer shares our opinion that it is common practice and well accepted not to confirm all results of a transcriptomics approach at the protein level.

Reviewer comment: Also, in Fig. 3E why was the 48h timepoint chosen? And why did the Authors look at variations of phosphor AKT after 72h from HOTAIR silencing (Fig. 4F). There is no explanation of the rationale behind the choice of these timepoints over others and why these differences between one experiment and another.

Response: We are sorry that the selection of the time points was not clearer explained in the manuscript. We now show measurements of all protein changes after 48h and 72h. As now can be appreciated much clearer, the kinetics of the protein changes differ. This is most probably because some of the changes are directly induced by HOTAIR, while others are not (as also indicated by the experiments shown in Supplementary Figure S3). We now also stress this point again in our revised manuscript on page 16 line 7: 'Most of these changes induced by HOTAIR silencing were already detectable 48 h after silencing (FGF8, Wnt inhibition, IL-12p35, CXCL12). However, phosphorylation of AKT and increased IL-6 secretion were only significant at later time points, suggesting indirect effects of decreased HOTAIR levels.'

Reviewer comment: Moreover, on page 14 the authors refer to IL12, but in the figure it's reported IL12A. So, why didn't they mention IL12A also in the main manuscript?

Response: We thank the reviewer for bringing this discrepancy to our attention and apologize for this error. The protein is IL-12p35 and the gene is IL-12A. This has been corrected in our revised manuscript page 16.

Reviewer comment: On the same page, it is totally unclear why CTNNB1, FGFR2 and LGR5 were chosen.

Response: We agree with the reviewer that our criteria to select specific genes for confirmation was not clearly stated and we now describe this clearer in the revised version. To study the mechanisms of HOTAIR regulation, we focused our analysis on the two main signaling pathways identified as regulated by HOTAIR i.e. FGF pathways and Wnt pathways and studied the genes significantly changed after HOTAIR silencing (p -value<0.05). This sentence is now included on page 15 line 20: 'For further confirmatory analysis at the mRNA and protein levels, we focused on the most changed transcripts within the most relevant enriched pathways.'

Reviewer comment: Fig. 6F and 6G, right sides: what is the difference between black and white dots? A proper legend is missing.

Response: We thank the reviewer for noting this mistake. The legend of Figure 7 (previously figure 6) has been changed to explain the color code of the dots. Black dots represent co-culture with SF, white dots with supernatants.

Reviewer comment: Fig. 7G: are the reported average values a combination of RA and OA samples? If so, why weren't they presented separately for the two different conditions?

Response: Indeed, Fig 7G represented a combination of RA and OA samples. As suggested by the reviewer and to improve the clarity of the findings, the two conditions are now presented separately for the two different conditions in the revised manuscript in Figure 8G (RA) and supplemental Figure S5 (OA).

Reviewer comment: Page 18: the conclusion that “could be an effect of decreased secretion of CXCL12 by HOTAIR silenced SF” is absolutely random and not supported by any experimental evidence.

Response: This conclusion has been removed from our result section in our revised manuscript.

Reviewer comment: Supplemental Table 2 appears cut (at least in the title) and cannot be accepted in the present format.

Response: Supplemental Tables S1 and S2 have been reformatted.

REVIEWERS' COMMENTS

Reviewer #1 (Remarks to the Author):

The authors have addressed the major issues.

Reviewer #2 (Remarks to the Author):

The Authors have addressed some of my concerns.

However, there are the following points that still need clarification:

1. If the silencing of a gene needs to be studied for its "pure" effect on a given phenotype, removing it completely is the best strategy. This can only be achieved by CRISPR/Cas9 deletion. The Authors argued that Gapmer is a better silencing method because it achieves a silencing closer to the one observed in an inflammatory environment. However, if Gapmer silences 95% after 2 days (as they replied to my comment 1) and in the inflammatory environment the silencing is 60% average (as they state in reply to my comment 2), their approach still does not recapitulate what happens in the inflammatory environment. Therefore, their approach continues to be unacceptable.
2. The HOTAIR 95% silencing during the new timecourse is not in Figure S4A (as stated in the rebuttal letter), but in S2A.
3. In the rebuttal letter, the Authors describe a significant p value if ≤ 0.08 . But in the statistical considerations, they wrote p is significant if ≤ 0.05 . At this point I'm very confused about the statistics in this manuscript.

We sincerely regret that there were inconsistencies in the revision of the manuscript and that we were not able to satisfactorily address all of the reviewer's points. Below you will find our point-by-point response.

Reviewers comment 1. *If the silencing of a gene needs to be studied for its "pure" effect on a given phenotype, removing it completely is the best strategy. This can only be achieved by CRISPR/Cas9 deletion. The Authors argued that Gapmer is a better silencing method because it achieves a silencing closer to the one observed in an inflammatory environment. However, if Gapmer silences 95% after 2 days (as they replied to my comment 1) and in the inflammatory environment the silencing is 60% average (as they state in reply to my comment 2), their approach still does not recapitulate what happens in the inflammatory environment. Therefore, their approach continues to be unacceptable.*

Answer: We unfortunately could not convince the reviewer of our point of view. However, we remain of the opinion that the partial transient knockdown of HOTAIR with GapmeR can provide relevant results for our hypothesis. We added the following sentence to the discussion: "However, we recognize that even with this approach, we are not able to fully recapitulate what actually happens in an inflammatory environment."

Reviewers comment 2. *The HOTAIR 95% silencing during the new timecourse is not in Figure S4A (as stated in the rebuttal letter), but in S2A.*

Answer: We apologize for this confusion in the rebuttal letter.

Reviewers comment 3. *In the rebuttal letter, the Authors describe a significant p value if ≤ 0.08 . But in the statistical considerations, they wrote p is significant if ≤ 0.05 . At this point I'm very confused about the statistics in this manuscript.*

Answer: We considered $p < 0.05$ as statistical significant throughout the manuscript. However, we still added $p \leq 0.08$ in the figures, because we think this still indicates that there is likely an effect.